# PROGRESSIVE MIX-UP FOR FEW-SHOT SUPERVISED MULTI-SOURCE DOMAIN TRANSFER

**Ronghang Zhu[1], Xiang Yu[2], Sheng Li[3,1]**
[1]University of Georgia, [2]Amazon, [3]University of Virginia
`ronghangzhu@uga.edu, yuxiang03@gmail.com, shengli@virginia.edu`

## ABSTRACT

This paper targets at a new and challenging setting of knowledge transfer from multiple source domains to a single target domain, where target data is few shot or even one shot with label. Traditional domain generalization or adaptation methods cannot directly work since there is no sufficient target domain distribution serving as the transfer object. The multi-source setting further prevents the transfer task as excessive domain gap introduced from all the source domains. To tackle this problem, we newly propose a progressive mix-up (P-Mixup) mechanism to introduce an intermediate mix-up domain, pushing both the source domains and the few-shot target domain aligned to this mix-up domain. Further by enforcing the mix-up domain to progressively move towards the source domains, we achieve the domain transfer from multi-source domains to the single one-shot target domain. Our P-Mixup is different from traditional mix-up that ours is with a progressive and adaptive mix-up ratio, following the curriculum learning spirit to better align the source and target domains. Moreover, our P-Mixup combines both pixel-level and feature-level mix-up to better enrich the data diversity. Experiments on two benchmarks show that our P-Mixup significantly outperforms the state-of-the-art methods, i.e., 6.0% and 8.6% improvements on Office-Home and DomainNet. Source code is available at `https://github.com/ronghangzhu/P-Mixup`

## 1 INTRODUCTION

Deep neural networks (DNN) have gained large achievements on a wide variety of computer vision tasks (He et al., 2016; Ren et al., 2015). As problems turn complex, the learned DNN models consistently fall short in generalizing to test data under different distributions from the training data. Such domain shift (Torralba & Efros, 2011) further results in performance degradation as models are overfitting to the training distributions. Domain adaptation (DA) (Xu et al., 2021; Zhu et al., 2021; Zhu & Li, 2021; Liu et al., 2023) has been extensively studied to address this challenge. Due to different settings regarding the source and target domains, DA problems vary into different categories such: unsupervised domain adaptation (UDA) (Zhu & Li, 2022a), supervised domain adaptation (SDA) (Motiian et al., 2017), and multi-source domain adaptation (MSDA) (Zhao et al., 2018). UDA aims to adopt knowledge from a fully labeled source domain to an unlabeled target domain. SDA intends to transfer knowledge from a fully labeled source domain to a partially labeled target domain. MSDA generalizes the UDA by adopting the knowledge from multiple fully labeled source domains to an unlabeled target domain. The main difficulty in the MSDA problem is how to achieve a meaningful alignment between the labeled source domains and the target domain that is unlabeled. Although DA has obtained some good achievements, assuming the availability of plenty of unlabeled/labeled target samples in real-world scenarios cannot be always guaranteed.

In this paper, we propose a challenging and realistic problem setting named Few-shot Supervised Multi-source Domain Transfer (FSMDT), by assuming that multiple labeled source domains are accessible but the target domain only contains few samples (i.e., one labeled sample per class), shown in Figure 1. Different from existing domain adaptation problems such as UDA, SDA and MSDA, the target domain in our problem does not provide any unlabeled samples to assist model training. The most relevant problem settings to ours are SDA and MSDA. SDA (Tzeng et al., 2015; Koniusz et al., 2017; Motiian et al., 2017; Morsing et al., 2021) seeks to transfer knowledge from a single source domain to a partially labeled target domain. The SDA methods cannot be simply used

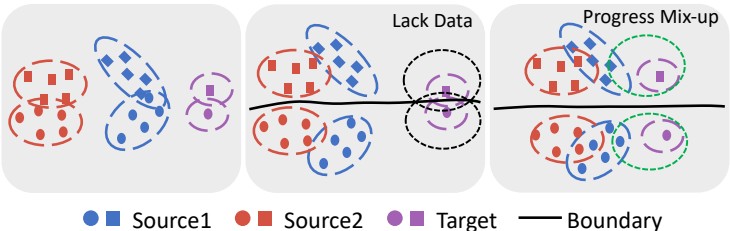

Figure 1: Visual illustration on the FSMDT problem (left), traditional domain adaptation solutions (middle) and our P-Mixup method (right).

to deal with our problem that involves multiple source domains, as the alignment among multiple source domains should be carefully addressed. In addition, existing MSDA methods (Duan et al., 2009; Sun et al., 2011; Zhao et al., 2018; Wang et al., 2020a; Zhou et al., 2021b; Ren et al., 2022) aim to learn domain-invariant representations by aligning the target domain to each of the source domains. However, these MSDA methods are not suitable for our FSMDT problem, as target domain only contains few labeled samples for training process which cannot support the domain invariance learning. Recently, multi-source few-shot domain adaptation (MFDA) (Yue et al., 2021) is proposed to address the application scenario where only a few samples in each source domain are annotated while the remaining source and target samples are unlabeled. Different from MFDA, our proposed FSMDT assumes only few target samples are available. The methods for MFDA would fail to learn discriminative representations on target domain in FSMDT due to insufficient target samples.

We propose a novel progressive mix-up scheme to tackle the challenges in the newly proposed FSMDT problem. Our scheme firstly creates an intermediate mix-up domain, which is initially set closer to the few-shot target domain. Rather than the commonly used image-level mix-up, we induce a cross-domain bi-level mix-up, which involves both the image-level mix-up and feature-level mix-up, to effectively enrich the data diversity. With the mix-up domain that is initially close to the target domain, the few-shot constraint on target domain is alleviated. Then, by enforcing the mix-up ratio to progressively favor towards the source domains, and meanwhile harnessing the target domain to be close to the mix-up domain, we gradually transfer knowledge from the multi-source domain to the target domain in a curriculum learning fashion. Furthermore, by optimizing over multiple source domains in a meta-learning regime, we present a stable and robust solution to the FSMDT problem.

Our main contributions are summarized as follows:

- We introduce a practical and challenging task, namely the Few-shot Supervised Multi-source Domain Transfer (FSMDT), which aims to transfer knowledge from multiple labeled source domains to a target domain with only few labeled samples.
- We propose a novel progressive mix-up scheme to help address the FSMDT problem, which creates an intermediate mix-up domain and gradually adapts the mix-up ratio to mitigate the domain shift between target domain and source domain.
- We conduct extensive experiments and show that our method successfully tackles the new FSMDT problem and it surpasses state-of-the-arts with large margins. In particular, it improves the accuracy by $6.0\%$ and $6.8\%$ over MSDA and SDA baselines on the Office-Home and DomainNet datasets, respectively.

## 2 RELATED WORK

### 2.1 DOMAIN ADAPTATION AND GENERALIZATION

Domain adaptation (DA) aims to transfer knowledge from a source domain to a target domain with a strong assumption that target data are available for model training. Domain generalization (DG) is a more challenging task in not only closing the domain gap but also addressing the absence of target data. There are two types of DA problems related to our proposed FSMDT problem: supervised domain adaptation (SDA) (Tzeng et al., 2015; Motiian et al., 2017; Morsing et al., 2021) and multi-source domain adaptation (MSDA) (Sun et al., 2011; Zhao et al., 2018; Wang et al., 2020a)

**Supervised Domain Adaptation** trains models by exploiting a partially labeled target domain and a single, fully labeled source domain. Seminal work such as the simultaneous deep transfer

(SDT) (Motiian et al., 2017) jointly learns domain-invariant features and aligns semantic information across domains by optimizing the domain confusion and distribution matching objectives. The classification and contrastive semantic alignment (CCSA) method (Motiian et al., 2017) uses the distribution alignment along the semantic manifold. To deal with the few-shot issue, CCSA reverts to point-wise surrogates of distribution and similarities. Recently, (Morsing et al., 2021) exploits graph embedding to encode intra-class and inter-class information to better align the source and target domains. Different from SDA, we consider multi-source domain instead of a single source domain, which is more challenging as real data is not constrained to be only from a distribution.

**Multi-Source Domain Adaptation** aims to learn domain-invariant feature across all domains, or leverage auxiliary classifiers trained with multi-source domain to ensemble a robust classifier for the target domain (Sun et al., 2011; Duan et al., 2009). ecently, the multi-source domain adversarial network (MDAN) (Zhao et al., 2018) theoretically analyzes the average case generalization bounds for MSDA classification and regression problems. In addition, the learning to combine for multi-source domain adaptation (LtC-MSDA) (Wang et al., 2020a) explores interactions among domains by building a knowledge graph of prototypes from various domains and investigates the information propagation among semantically adjacent representations. Despite the good performance, none of the above methods consider the practical scenario with only very few labeled target samples.

**Domain Generalization** aims to learn a model from multiple source domains that can generalize well on unseen target domain. Existed DG methods can be roughly divided into three groups. Domain alignment based methods (Muandet et al., 2013; Li et al., 2018b) aim to learn the domain invariant features by aligning feature distributions across multiple source domains. Meta-learning based methods (Li et al., 2018a; Shu et al., 2021) divide multiple source domains into the meta-train and meta-test sets, and learn a model on the meta-train set with the intention of improving its performance on the meta-test set. Data augmentation based methods (Zhou et al., 2021a; 2020; Zhu & Li, 2022b) aim to improve the generalization of learned models by enriching the diversity of source domains. Though domain generalization addresses the unseen target domain, which is a harder problem than our few-shot seen target setting, it is not suitable for our FSMDT problem as it doesn't consider how to utilize these available few-shot samples in target domain.

## 2.2 DATA AUGMENTATION BY MIX-UP

Mix-up (Zhang et al., 2018) is a data augmentation technique that has been widely applied in self-supervised learning, domain adaptation, and domain generalization. Dual mixup regularized learning (DMRL) (Wu et al., 2020) conducts class-level and domain-level mix-up strategies to learn a domain-invariant feature space. Recently, Domain-augmented meta learning (DAML) (Shu et al., 2021) applies multi-source mix-up strategy to augment source domains. However, most methods interpolate samples with a pre-defined mix-up ratio distribution, e.g., beta distribution. Lately, MetaMixup (Mai et al., 2021) proposes a meta-learning based framework to dynamically update mix-up ratio. However, it requires a special validation setting to learn the mix-up ratio, and it does not consider the mix-up problems across multiple domains. In contrast, we consider the cross-domain mix-up and propose a progressive mix-up scheme based on the cross-domain Wasserstein distance, which does not rely on extra validation settings.

## 2.3 FEW-SHOT LEARNING

Few-shot learning (Wang et al., 2020b) aims to learn a model that can be easily adapted to novel tasks with limited labeled data. To tackle this challenging problem, plenty of methods have been proposed which can be roughly divided into metric learning based method (Snell et al., 2017; Sung et al., 2018), meta-learning based method Finn et al. (2017); Chen et al. (2021), optimization based method (Lee et al., 2019; Ravi & Larochelle, 2017), and data augmentation based method (Li et al., 2020; Xu & Le, 2022). Snell et al. (2017) proposes the Prototypical Network (PTN) to learn a metric space for classification. Finn et al. (2017) designs a Model-Agnostic Meta-Learning (MAML) framework which can learn the model under various tasks, such that it can be easily adopted to novel tasks with a few labeled data. Li et al. (2020) proposes a conditional Wasserstein generative adversarial networks based adversarial feature generator to enrich the diversity of the available limited data for novel tasks. Recently, a benchmark, namely Meta-Dataset (Triantafillou et al., 2020), is proposed for Multi-Domain Few-Shot Learning (MDFS) problem (Dvornik et al., 2020; Liu et al.,

2021). One of the state-of-the-art methods, Universal Representation Transformer (URT) (Liu et al., 2021) is designed to transfer the learned universal representation to task-specific representation. Even though MDFS is very similar to our proposed problem, there still existing significant difference between MDFS and our FSMDT, e.g., our proposed FSMDT assumes that the target label space is contained in multi-source label space while MDFS holds the assumption that the target label space is excluded in multi-source label space.

## 3 METHOD

Unlike SDA, we target at jointly leveraging multi-source domain other than single source domain, together with few-shot labeled target samples, to adapt the multi-source domain knowledge to the target domain. The main challenge is the extremely limited target data points, which cannot provide sufficient and stable target distribution and thus difficult to conduct transfer. Inspired by Mix-up (Zhang et al., 2018), we propose a progressive mix-up (P-Mix) scheme to introduce an intermediate mix-up domain, and enforce the distribution alignment of "source to mix-up" and "target to mix-up". Our scheme starts with a mix-up distribution close to the target domain, and gradually drifts towards source domains. In this way, the large domain gap is surrogated by a milder intermediate gap and the target to source alignment is indirectly achieved. Firstly, we introduce the preliminary. Then, we give details of the bi-level mix-up. Last, we illustrate our newly proposed progressive mix-up scheme and summarize the overall pipeline of our algorithm.

### 3.1 PRELIMINARIES

In Few-shot Supervised multi-source domain Transfer (FSMDT) problem, we have $M$ full labeled source domains and a target domain with few-shot labeled data. The $i$-th source domain $\mathcal{D}_{s,i} = \{(x_{s,i}^j, y_{s,i}^j)\}_{j=1}^{N_{s,i}}$ contains $N_{s,i}$ labeled samples drawn from the source distribution $P_{s,i}(x,y)$, and the target domain $D_t = \{(x_t^j, y_t^j)\}_{j=1}^{N_t}$ includes $N_t$ labeled samples selected from the target distribution $P_t(x,y)$. Here, $N_t \ll N_{s,i}$, i.e., $N_t$ can be as few as 1-shot per class. $P_t(x,y) \neq P_{s,i}(x,y)$, and $P_{s,i}(x,y) \neq P_{s,j}(x,y)$ where $i \neq j$. The multiple source domains and target domain have the same label space $Y = \{1, 2, \ldots, K\}$ with $K$ categories. We aim to learn an adaptive model $\mathbf{H}$ on $\{\mathcal{D}_{s,i}\}_{i=1}^M$ and $\mathcal{D}_t$, that can generalize well on unseen samples from target domain. In general, $\mathbf{H}$ consists of two functions, i.e., $\mathbf{H} = \mathbf{F} \circ \mathbf{G}$. Here $\mathbf{G} : x \to g$ represents the feature extractor that maps the input sample $x$ into an embedding space, and $\mathbf{F} : g \to f$ is the classifier with input the embedding to predict the category.

#### 3.1.1 RECAP OF MIX-UP

Mix-up (Zhang et al., 2018) is one of the most popular data augmentation strategies to improve the generalization of the learned model by enriching the diversity of the original domain. The core idea of mix-up is to create virtual samples by randomly interpolating two samples in a convex fashion. Specifically, given two samples $(x_i, y_i)$ and $(x_j, y_j)$, the virtual sample $(\tilde{x}, \tilde{y})$ is defined as:

$$\tilde{x} = \lambda x_i + (1 - \lambda)x_j, \tag{1}$$
$$\tilde{y} = \lambda y_i + (1 - \lambda)y_j, \tag{2}$$

where label $y$ is the one-hot label encoding and $\lambda$ is randomly sampled from a predefined distribution, e.g., beta distribution.

### 3.2 CROSS-DOMAIN BI-LEVEL MIX-UP

Traditional mix-up is originally designed for self-supervised learning, i.e., introducing a new class data by interpolating from two known classes' data, which can increase the training data diversity. When considering the domain transfer problem, such mix-up will be cross-domain, i.e., a data point from source domain and a data point from target domain. Meanwhile, besides the pixel-level mix-up, recent manifold based mix-up (Verma et al., 2019; Shu et al., 2021; Xu et al., 2020) shows that feature-level interpolation can also improve the generalization and model robustness. We thus investigate both the pixel-level and feature-level mix-ups.

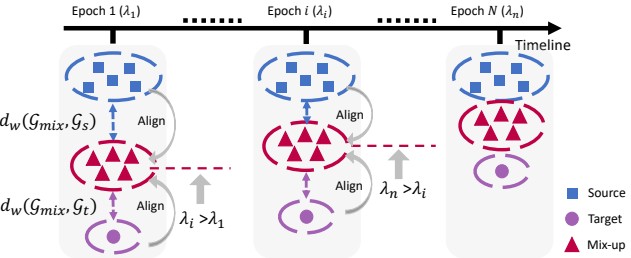

Figure 2: The flowchart of the proposed progressive mix-up. A mix-up domain (red) is introduced as initially closer to the target domain. By enforcing the mix-up ratio $\lambda$ to be progressively increasing based on the wasserstein distance of source-to-mixup and target-to-mixup, we push the mix-up domain gradually to be closer to source domains, and thus achieving the alignment of multi-source domain to the few-shot target domain.

**Cross-Domain Image-Level Mix-up.** Motivated by the success of mix-up in self-supervised learning, we apply it to our domain transfer task, which can create new samples with new labels. We utilize it to largely enrich the target domain distribution as there are overly limited target samples. The source and target samples are linearly interpolated as:

$$\tilde{x}_{img} = \lambda x_{s,i} + (1 - \lambda)x_t, \tag{3}$$
$$\tilde{y}_{img} = \lambda y_{s,i} + (1 - \lambda)y_t, \tag{4}$$

where $\lambda$ is the mix-up ratio. Notice that during training, such mix-up ratio can be adjusted, e.g., a larger $\lambda$ generates closer-to-source samples and a smaller $\lambda$ generates closer-to-target samples.

**Cross-Domain Feature-Level Mix-up.** On the learned feature representation manifold, mix-up at the feature level enables more intermediate virtual features to increase the feature diversity and can directly interact with the classifier $\mathbf{F}$ learning. Here, given a pair of source and target features and their corresponding labels: $(g_{s,i}, y_{s_i})$ and $(g_t, y_t)$, we have

$$\tilde{g}_{feat} = \lambda g_{s,i} + (1 - \lambda)g_t, \tag{5}$$
$$\tilde{y}_{feat} = \lambda y_{s,i} + (1 - \lambda)y_t, \tag{6}$$

where $\lambda$ is the mix-up ratio same as the one used in image-level mix-up. With exactly the same $\lambda$, we argue that the image-level mix-up samples lie in the same feature space as the feature-level mix-up samples. Thus, we can jointly utilize the two for penalty, i.e., the same class image-level mix-up and feature-level mix-up should go for the same classification result.

### 3.3 PROGRESSIVE MIX-UP SCHEME

Previous work apply either fixed sampling or some simple randomized sampling for the mix-up ratio $\lambda$, e.g., beta or dirichlet distribution (Zhang et al., 2018; Wu et al., 2020; Xu et al., 2020; Shu et al., 2021). However, we find that the sampling of mix-up ratio is crucial for the domain transfer. The ratio directly determines the intermediate mix-up domain. If a mix-up domain is constant or some special distribution, the alignment is either still constantly hard or likely to be under-fitting, supported from a recent work MetaMixup (Mai et al., 2021).

To alleviate it, we dig into the Wasserstein distance of "source-to-mixup" $d_w(\mathcal{G}_s, \mathcal{G}_{mix})$ and "target-to-mixup" $d_w(\mathcal{G}_t, \mathcal{G}_{mix})$, where $\mathcal{G}_s, \mathcal{G}_t, \mathcal{G}_{mix}$ stand for the embeddings of source, target and mix-up domains. We observe that during the training, if the mix-up domain initially is closer to the few-shot target domain, the alignment is relatively simple as $d_w(\mathcal{G}_t, \mathcal{G}_{mix})$ is already small while $d_w(\mathcal{G}_s, \mathcal{G}_{mix})$ can be effectively minimized as there are sufficient source domain data. When gradually increasing the mix-up ratio towards closer to source domains, since we already harness the "target-to-mixup" distance to be small, we are pushing the entire mix-up domain and few-shot target domain towards the source domains, as illustrated in Figure 2. Such progressively adjusted mix-up ratio, following the spirit of curriculum learning (Bengio et al., 2009), eases the initial large domain gap by mildly starting close to the target, and secures the entire transfer process smoothly.

Specifically, we introduce a weighting factor $q$ to depict the closeness to source as:

$$q = \exp\left(-\frac{d_w(\mathcal{G}_s, \mathcal{G}_{mix})}{(d_w(\mathcal{G}_s, \mathcal{G}_{mix}) + d_w(\mathcal{G}_t, \mathcal{G}_{mix}))T}\right), \tag{7}$$

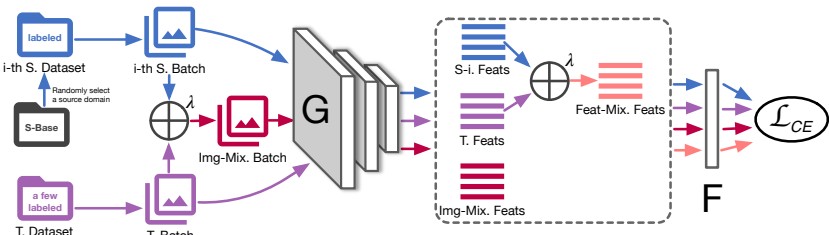

Figure 3: The training architecture. Both image and feature level P-Mixup are applied for the cross-entropy loss. **G** is the feature extractor and **F** is the classifier.

where $T$ is a temperature factor defined as 0.05. During training, by initializing $\mathcal{G}_{mix}$ closer to target domain, such $q$ is small. To progressively adjust it, we consider to apply this closeness on top of the previous stage $\lambda$ in a moving average manner. Further, a linearly incremental component is introduced to enforce the gradual closeness to the source domains. The progressive mix-up is formulated as:

$$\lambda_n = \frac{n(1-q)}{N} + q\lambda_{n-1}, \tag{8}$$

where $N$ is the total number of iterations and $n$ is the current iteration index. Initial weighting $\lambda_0$ towards source is 0. To numerically stabilize the training procedure, we introduce a uniform distribution $U$, a random perturbation on top of the current $\lambda_n$:

$$\tilde{\lambda}_n = \text{Clamp}(U(\lambda_n - \sigma, \lambda_n + \sigma), \min = 0.0, \max = 1.0), \tag{9}$$

where $\sigma$ is a local perturbation range, i.e., we empirically set it as 0.2. $\tilde{\lambda}_n$ is then stochastically sampled and clamped into range $[0.0, 1.0]$ for each iteration $n$'s mix-up ratio.

### 3.4 ARCHITECTURE AND LEARNING OBJECTIVES

The overall architecture is shown in Figure 3, which mainly consists of a feature generator **G** and a classifier **F**. During training, since there are multiple labeled source domain data, and a single few-shot target domain data, we follow the canonical domain generalization frameworks such as MAML (Finn et al., 2017), to organize our training in a meta-learning manner.

Denoting the model parameters $\mathbf{F} \circ \mathbf{G}$ as $\theta$, the objective for classification is defined as:

$$\mathcal{L}_{ce}^{\mathcal{T}_i}(\theta) = -\sum_{x,y\in\mathcal{T}_i}\sum_{k=1}^{K} y_k \log(\theta(x)_k), \tag{10}$$

where $\mathcal{T}_i$ stands for a specific domain, e.g., one of the source domains or the target domain, $x$ is the input image and $y$ is the ground truth label, and $K$ is the number of classes. Notice that for "cross domain image-level mix-up", the input is the mix-up image $\tilde{x}_{img}$ and the label is the mix-up label $\tilde{y}_{img}$. For "cross domain feature-level mix-up", the mix-up feature $\tilde{g}_{feat}$ is fed into the classifier and computes the $\mathcal{L}_{ce}$ loss. Meanwhile, the label $y$ is the mix-up label $\tilde{y}_{feat}$.

Following MAML, we conduct a meta-optimization to pseudo-update the model parameters for the first time by minimizing $\sum_{\mathcal{T}_i\in p(\mathcal{T})} \mathcal{L}_{ce}^{\mathcal{T}_i}$:

$$\theta' = \theta - \alpha\nabla_\theta \sum_{\mathcal{T}_i\in p(\mathcal{T})} \mathcal{L}_{ce}^{\mathcal{T}_i}(\theta). \tag{11}$$

$p(\mathcal{T})$ is a sampling distribution among the meta-train domains, e.g., each of the three benchmarks in our experiments contains four domains, we uniformly sample two out of three source domains. The remaining source, target and the mix-up domains are used for meta-test to update the model as:

$$\theta = \theta - \beta\nabla_\theta \sum_{\mathcal{T}_j\notin p(\mathcal{T})} \mathcal{L}_{ce}^{\mathcal{T}_j}(\theta'), \tag{12}$$

where $\alpha$ and $\beta$ are the update step size for meta-train and meta-test respectively. To simplify the parameters, we set $\alpha = \beta = 0.001$. Notice that the mix-up domain contains two sub-domains, the image-level mix-up and the feature-level mix-up. Both of them are used in meta-test.

Table 1: mAP(%) on Office-Home. Named in row is the target domain which contains 10 classes randomly selected from label space. (A: Art, C: Clipart, P: Product, R: Real_world)

| $\mathcal{D}_t^{10}$ | ERM-w/o | ERM-w | CCSA | MDAN | Mix-up | DAML | URT | Ours |
|---|---|---|---|---|---|---|---|---|
| A | 60.48 | 60.83 | 64.52 | 58.49 | 59.50 | 60.30 | 59.48 | **72.23** |
| C | 44.06 | 46.35 | 56.32 | 44.17 | 51.03 | 49.88 | 52.41 | **59.97** |
| P | 72.55 | 72.75 | 75.89 | 70.43 | 73.03 | 72.96 | 82.11 | **82.69** |
| R | 79.32 | 76.40 | 79.17 | 74.28 | 76.91 | 76.17 | 81.52 | **85.05** |
| Ave. | 63.83 | 64.08 | 68.97 | 61.84 | 65.12 | 64.83 | 68.88 | **74.99** |

## 4 EXPERIMENT

### 4.1 EXPERIMENTAL SETTINGS

**Datasets:** We adopt two standard domain adaptation and generalization benchmarks: **(1) Office-Home** (Venkateswara et al., 2017) which consists of four domains (Art, Clipart, Produce, and Real_world) with 65 classes. **(2) DomainNet** (Peng et al., 2019) contains 345 classes. We conduct experiments on four domains (Clipart, Painting, Real, and Sketch) from it.

**Protocols:** To highlight the challenging few-shot target domain setting, we cannot anymore use the original protocols from the above two datasets. We observe that even with one-shot, since the number of classes are many, e.g., 345 classes from DomainNet, utilizing all the classes can provide a sufficient diversified target domain distribution. To exactly constrain the target distribution to be few-shot, for Office-Home, we randomly select 10 out of 65 classes each with one sample as the target. Similarly, we randomly select 15 out of 345 classes for DomainNet. The remaining samples in these selected classes are used as the test data. Such random sampling is conducted for 5 times and the averaged result is reported.

**Baselines:** We compare with four main streams of methods: (1) *Multi-Source Few Shot Learning* method, namely Universal Representation Transformer (URT) (Liu et al., 2021).(2) *Supervised Domain Adaptation* method, namely Classification and Contrastive Semantic Alignment (CCSA) (Motiian et al., 2017). (3) *Multi-Source Domain Adaptation*, namely Multisource Domain Adversarial Networks (MDAN) (Zhao et al., 2018). (4) *Domain Generalization* method, namely Domain-Augmented Meta Learning (DAML) (Shu et al., 2021). (5) *Data Augmentation* method, namely Mix-up (Zhang et al., 2018). Besides, we consider another general baseline, i.e., Empirical Risk Minimization (Koltchinskii, 2011) with/without labeled target domain (ERM-w, ERM-w/o).

**Evaluation Metrics:** For each of the benchmarks, each domain is in turn regarded as the target domain while the remaining are considered as source domains. For each experiment, we report the mean average precision (mAP) by averaging over 5 times of all the class' average precision. We fix the random seed to 1-5 when self-constructing the new domain and sampling target samples so the results of different methods can be fairly compared.

**Implementation Details:** Our implementation is based on Pytorch (Paszke et al., 2019). We use ResNet-18 (He et al., 2016) pretrained on ImageNet (Deng et al., 2009) as the backbone network. We optimize the model using SGD with momentum of 0.9 and weight decay of $5 \times 10^{-4}$. The batch size is set to 50. The initial learning rate is set to 0.001. For all the compared methods and Ours, we use the same basic data preprocessing on the image and the same backbone.

### 4.2 MAIN RESULTS

**Office-Home:** In Table 1, comparing ERM-w to ERM-w/o, we observe that the labeled target domain containing 10 images cannot directly improve the performance, which verifies the setting is indeed challenging. Third column is the representative supervised domain adaptation method, CCSA, clearly outperforms the baseline ERM-W. There is also MDAN in the fourth column whose performance is worse than ERM-W, as there is no sufficient target distribution to support the adaptation. Across all the methods, our approach demonstrates clear advantages, i.e., when compared to the second best, CCSA, the gain is as significant as 6.02% on "Ave."

**DomainNet:** As shown in Table 2, our method's performance in both 10 labeled target samples and 15 labeled target samples scenarios show a clear advantage over all the baselines. Compared to the most competitive opponent URT, our method gets the best performance on three out of four

Table 2: mAP(%) on DomainNet. Named in column is the target domain contains 10/15 classes randomly selected from label space. (C: Clipart, P: Painting, R: Real, S: Sketch)

| Method | $\mathcal{D}_t^{10}$ | | | | | $\mathcal{D}_t^{15}$ | | | | |
|--------|-------|-------|-------|-------|-------|-------|-------|-------|-------|-------|
| | C | P | R | S | Avg. | C | P | R | S | Avg. |
| ERM-w/o | 53.58 | 48.03 | 57.70 | 45.57 | 51.22 | 51.90 | 45.68 | 57.11 | 43.51 | 49.55 |
| ERM-w | 54.91 | 48.39 | 58.27 | 48.27 | 52.46 | 54.50 | 47.51 | 58.23 | 45.17 | 51.35 |
| CCSA | 58.78 | 54.40 | 61.32 | 56.74 | 57.81 | 53.51 | 50.90 | 58.63 | 52.11 | 53.79 |
| MDAN | 56.07 | 48.50 | 59.32 | 47.40 | 52.82 | 54.98 | 46.95 | 59.24 | 45.36 | 51.63 |
| Mix-up | 61.81 | 59.49 | 66.41 | 56.17 | 60.97 | 56.85 | 53.98 | 64.51 | 50.25 | 56.40 |
| DAML | 58.26 | 49.33 | 55.53 | 47.15 | 52.57 | 56.56 | 47.63 | 57.39 | 46.45 | 52.01 |
| URT | 67.18 | 56.87 | **84.20** | 55.08 | 65.83 | 53.07 | 47.99 | 72.53 | 41.00 | 53.65 |
| Ours | **79.32** | **70.18** | 82.63 | **69.88** | **75.50** | **72.46** | **64.37** | **76.88** | **60.41** | **68.53** |

Table 3: Ablation study on Office-Home. Named in column is the target domain which contains 10 classes randomly selected from the label space.

| Mix-up Ratio $\lambda$ | Method | Art | Clipart | Product | Real_world | Ave. |
|------------------------|--------|-----|---------|---------|------------|------|
| N/A | ERM-w (no mix-up) | 60.48 | 44.06 | 72.55 | 79.32 | 63.83 |
| Random Sampling | Feat-Mix | 60.06 | 48.21 | 69.06 | 73.56 | 62.72 |
| | Img-Mix | 59.50 | 51.03 | 73.03 | 76.91 | 65.12 |
| | Feat-Mix + Img-Mix | 66.40 | 56.18 | 76.51 | 78.98 | 69.52 |
| Progressive Update | Feat-Mix | 64.89 | 55.08 | 75.41 | 77.52 | 68.22 |
| | Img-Mix | 68.45 | 57.55 | 81.26 | 83.89 | 72.79 |
| | Feat-Mix + Img-Mix | 72.23 | 59.97 | 82.69 | 85.05 | 74.99 |

tasks and surpasses by $9.67\%$ on 10 labeled target samples "Ave." and $14.88\%$ on 15 labeled target samples "Ave.". In this dataset, we find that ERM-w obtains the same level performance as most of baselines, e.g., supervised domain adaptation method CCSA, partially showing that this benchmark is more challenging as the domain gap becomes more challenging compared to the other two datasets. Overall, these results strongly demonstrate the effectiveness of our proposed progressive mix-up for improving domain transfer with extremely few labeled target domain samples.

## 4.3 ABLATION STUDY

We conduct a comprehensive ablation study to examine the effectiveness of our proposed core components in Table 3. The baseline of ERM-w utilizing the target domain data but without mix-up is shown in the first row. Feat-Mix denotes the cross-domain feature-level mix-up and Img-Mix indicates the cross-domain image-level mix-up. We introduce the general mix-up ratio sampling strategy ($\lambda \sim Beta(0, 1)$) used in Mix-up (Zhang et al., 2018), as a major comparison. Firstly, We observe that the bottom row methods consistently outperform the middle row by a large margin, highlighting the superiority of the proposed progressive mix-up strategy. Then, we look into the combination of modules within each sampling method. We observe that cross-domain image-level mix-up (Img-Mix) shows better result over cross-domain feature-level mix-up (Feat-Mix) with more than $4.0\%$ "Ave." improvement. If going for one module, image-level mix-up would be a better choice. If with no restriction, a combination of both image and feature level mix-ups can further boost the accuracy, because a combined mix-up enriches the data diversity more than each of the single choices.

## 4.4 EFFECT OF TARGET SAMPLE SHOTS

We investigate the effect of the number of target sample shots on our proposed P-Mixup with DomainNet where "Clipart" is selected as the target domain. We increase the number of selected classes from 10 to 345. Corresponding, the number of available target samples range from 10 to 345. As shown in Table 4, our method's performance across different numbers of selected classes settings show a clear advantage over all the baselines. Specifically, even when we reach the full classes, i.e., 345, our method still surpasses the most competitive opponent DAML, by $3.27\%$. As the size of

Table 4: mAP($\%$) on DomainNet "Clipart" setting where "Clipart" is the target domain. $n$ indicates the number of classes randomly selected from the label space.

| $\mathcal{D}_t^n$ | ERM-w/o | ERM-w | CCSA | MDAN | Mix-up | DAML | URT | Ours |
|---|---|---|---|---|---|---|---|---|
| 10 | 53.58 | 54.91 | 58.78 | 56.07 | 61.81 | 58.26 | 67.18 | 79.32 |
| 15 | 51.90 | 54.50 | 53.51 | 54.98 | 56.85 | 56.56 | 53.07 | 72.46 |
| 20 | 57.57 | 59.23 | 56.48 | 58.50 | 60.16 | 58.22 | 53.67 | 70.16 |
| 25 | 54.45 | 55.71 | 45.43 | 54.55 | 50.54 | 57.82 | 59.03 | 68.74 |
| 100 | 55.68 | 57.16 | 51.98 | 58.74 | 46.69 | 59.25 | 28.27 | 63.48 |
| 345 | 55.94 | 57.86 | 48.73 | 57.47 | 46.19 | 60.36 | 20.48 | 63.63 |

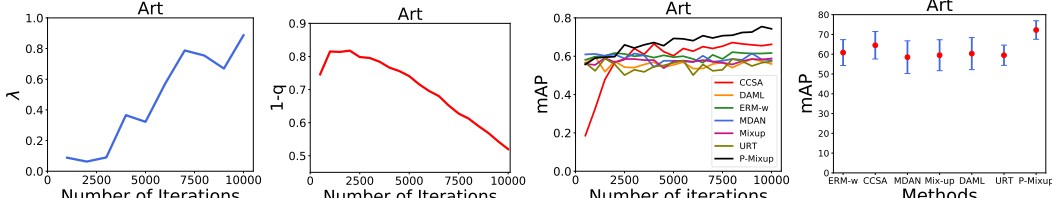

Figure 4: As shown from left to right, the first two figures illustrate the Mix-up ratio $\lambda$ and the $q$ value introduced in Equation 7. The lower $1 - q$, the closer the target is to the sources. The last two figures describe the P-Mixup training behavior compared to baselines and standard deviation (STD) values for all methods.

available labeled target samples decreases, our method still holds an obvious advantage, which further confirms that our method is more advantageous when target sample shots are extremely fewer.

## 4.5 Mix-up Ratio and Computation Analysis

As shown the first subfigure in Figure 4, we validate the P-Mixup scheme by showing the mix-up ratio and the $q$ value introduced in Equation 7 on Office-Home. The subfigure shows the trend of the proposed mix-up ratio $\lambda$ along the training iterations. Generally it is an increasing tread as we gradually push the mix-up domain to be closer to source domains. The second subfigure in Figure 4 shows $1 - q$ over iterations, which indicates the distance change between source and target domains. As $q$ depicts the closeness to source, we use $1 - q$ to present the closeness to target. During the first 4000 iteration, the mix-up distribution is closer to target than source, and the model gradually handles the"target-to-mixup" distance to be small. As a result, we observe that the $1 - q$ value gently turns small. After the model harnesses the "target-to-mixup" distance, the mix-up distribution gradually moves close to source domains as $\lambda$ goes up. Afterwards, the "target-to-mixup" distance continually decreases, showing that the source domains are continuously transferred onto the target domain and our P-Mixup is indeed effective in mitigating the domain shift in FSMDT. The last two subfigures in Figure 4 show the training behavior and standard deviation (STD) values for all methods on Office-Home. We observe that our proposed method P-Mixup consistently and significantly outperforms all the baselines in terms of training behavior and STD, which verify the effectiveness of our P-Mixup.

## 5 Conclusions

In this work, we propose to address a new and challenging problem, namely Few-shot Supervised multi-source domain Transfer (FSMDT), where multiple fully labeled source domain samples and extremely limited target samples are accessible. A progressive mix-up (P-Mixup) scheme is newly introduced to effectively mitigate the source and target domain gap especially when the target domain is with extremely few-shot samples. We jointly consider the image-level and feature-level cross-domain mix-up to sufficiently enrich the data diversity. A meta-learning optimization strategy is applied to support the multi-domain joint training with stable and robust convergence. Extensive experiments show that our method achieves significant performance gain over the state-of-the-art methods across two main domain adaptation benchmarks.

ACKNOWLEDGEMENT

This research is supported by the the U.S. Army Research Office Award under Grant Number W911NF-21-1-0109 and the Cisco Faculty Award.

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

## A  APPENDIX

The appendix provides additional experiments and justifications of the proposed progressive mix-up (P-Mixup) method. In the following sections, we firstly introduce details of benchmark datasets. Then, we show the specific process of implementation and present the analysis of the hyper-parameter sensitivity study on $\sigma$ used in our method. Next, we investigate the influence of moving direction of mix-up distribution on proposed P-Mixup. Furthermore, we provide the standard deviation (STD) values and training behavior for all the methods across all the benchmark datasets. Finally, we analyse the limitation of our method and provide insights on potential directions for furture research.

The mix-up ratio update formula in our method is defined as:

$$\tilde{\lambda}_n = \text{Clamp}(U(\lambda_n - \sigma, \lambda_n + \sigma), \min = 0.0, \max = 1.0), \tag{13}$$

where $\lambda_n$ is the progressive mix-up ratio at $n$-th iteration. $U$ is a uniform distribution with a local perturbation range $\sigma$.

### A.1  DATASETS

Table 5 shows the overall descriptions of benchmark datasets, i.e., **PACS** (Li et al., 2017), **Office-Home** (Venkateswara et al., 2017) and **DomainNet** (Peng et al., 2019). (1) **PACS** is a recent challenging domain adaptation/generalization benchmark. It consists of seven object categories from four domains, namely art paintings, cartoon, sketch, and photo. (2) **Office-Home** is a more challenging dataset than **PACS** as it has a large domain gap, which includes 15,588 images from 65 categories in office and home circumstance, consisted by four particularly dissimilar domains: Artistic images, Clip Art, Product images, and Real-World images. (3) **DomainNet** is the largest domain adaptation dataset which includes 345 classes and six domains (Clipart, Infograph, Painting, Quickdraw, Real, and Sketch). In our setting, we conduct experiments on four domains of Clipart, Painting, Real, and Sketch.

Table 5: Statistics of the three benchmark datasets.

| Datasets | Domains/Samples | | | | Classes |
|---|---|---|---|---|---|
| **PACS** | Art painting/2,048 | Cartoon/2,344 | Photo/1,670 | Sketch/3,929 | 7 |
| **Office-Home** | Art/2,427 | Clipart/4,365 | Product/4,439 | Real_world/4,357 | 65 |
| **DomainNet** | Clipart/48,129 | Painting/72,266 | Real/172,947 | Sketch/69,128 | 345 |

### A.2  IMPLEMENTATION DETAILS

We implement our P-Mixup in Pytorch (Paszke et al., 2019). We adopt the ImageNet pre-trained ResNet-18 (He et al., 2016) as the feature extractor **G** and optimize it with SGD as the optimization algorithm. We train the model for 10,000 iterations on **Office-Home**, and 30,000 iterations on **DomainNet**. We update the mix-up ratio $\lambda_n$ every 100 iterations. Following the idea of MAML (Finn et al., 2017), for each 3 iterations, we randomly select 2 source domains as the meta-train domain, and the rest source, target, and mixup domains are meta-test domain. The first 2 iterations is meta-train, and the last iteration is meta-test that contains source, target, and their bi-level mixed samples. The learning rates $\alpha$ and $\beta$ are set to 0.01. For all the baselines, we use the same basic image processing procedures and the same feature extractor as our P-Mixup.

### A.3  SENSITIVITY STUDY ON $\sigma$

To analyze the sensitivity of our P-Mixup to the hyper-parameter $\sigma$, we conduct experiments on Office-Home for all four protocols. The value $\sigma$ is selected from $\{0.05, 0.10, 0.15, 0.20, 0.25, 0.30, 0.35, 0.40\}$. As shown in Table 6, we observe that the performance of P-Mixup slightly increases in the range $[0.05, 0.25]$ when the value of $\sigma$ is increased, and then the performance is relatively stable in the range $[0.25, 0.40]$. Overall, our P-Mixup is not sensitive to the value of $\sigma$.

Table 6: Impact of $\sigma$ in our P-Mixup on Office-Home (averaged over 5 times).

| $\sigma$ in Eq. 13 | Art | Clipart | Product | Real_world | Ave. |
|---|---|---|---|---|---|
| 0.05 | 70.06 | 60.19 | 82.14 | 84.57 | 74.24 |
| 0.10 | 70.42 | 60.55 | 82.62 | 85.18 | 74.69 |
| 0.15 | 70.63 | 61.04 | 83.45 | 85.72 | 75.21 |
| 0.20 | 72.23 | 59.97 | 82.69 | 85.05 | 74.99 |
| 0.25 | 71.83 | 61.50 | 83.91 | 86.42 | 75.92 |
| 0.30 | 71.71 | 60.69 | 84.35 | 86.62 | 75.84 |
| 0.35 | 72.07 | 60.75 | 83.83 | 86.19 | 75.71 |
| 0.40 | 72.46 | 60.61 | 84.04 | 86.52 | 75.91 |

Table 7: mAP(%) on PACS. Named in row is the target domain which contains 4 classes randomly selected from label space. Rest domains are source domains.

| $\mathcal{D}_t^4$ | ERM-w/o | ERM-w | CCSA | MDAN | Mix-up | DAML | URT | Ours |
|---|---|---|---|---|---|---|---|---|
| A | 76.42 | 78.47 | 75.39 | 77.84 | 79.84 | 79.37 | 59.36 | **83.21** |
| C | 70.15 | 72.37 | 74.04 | 70.15 | 71.57 | 76.86 | 62.61 | **79.96** |
| P | 86.79 | 89.95 | 87.24 | 91.16 | 93.75 | 91.79 | 79.40 | **95.62** |
| S | 64.98 | 69.22 | 73.29 | 70.10 | 66.32 | 72.79 | 58.71 | **81.97** |
| Ave. | 74.58 | 77.50 | 77.49 | 77.31 | 77.87 | 80.20 | 65.02 | **85.19** |

## A.4  RESULTS ON PACS

Evaluation on PACS is shown in Table 7. We observe that our method consistently and significantly outperforms all the baselines. Specifically, we have $7.69\%$ "Ave." performance gain compared with ERM-w and $4.99\%$ "Ave." improvement compared with the second best DAML. A side observation is that all the baselines obtain relatively better performance compared to Office-Home and Doman-Net datasets, which suggests that the PACS could be less challenging, as 4 images from target domain could notably boost the performance, i.e., when comparing to ERM-w/o.

## A.5  META-LEARNING ABLATION STUDY

We investigate the behavior of our method on different meta-train and meta-test splittings on Office-Home with 10-shot. As the Office-Home dataset contains 4 domains, for each task, there are 3 domains are selected as the source domains and the remaining is the target domain. We also have the mix-up domain in each task. Due to the limited target data, we simplify the splitting by treating the target and mix-up domains as the whole denoted as $\mathcal{D}_{mix-up}$. As shown in Table 8, we increase the size of meta-train set from 1 source domain to 3 source domains. Corresponding, the remaining domains are adopted as the meta-test set. We find that different meta-learning splittings achieve the similar performance, and the meta-train set with three source domains slightly outperforms others.

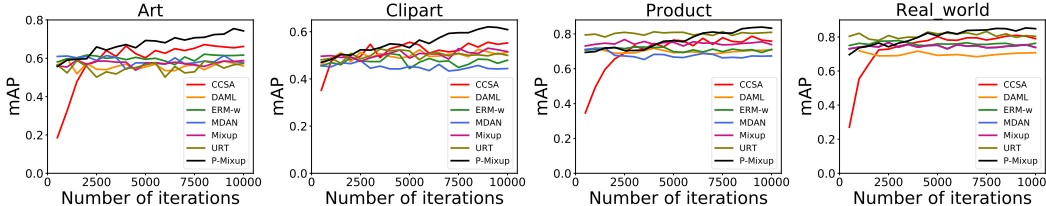

Figure 5: Illustration of our P-Mixup training behavior compared to other baselines on Office-Home all four protocols, where the one in caption is selected as the target domain and the rest are source domains.

Table 8: Ablation study on Office-Home with 10-shot under different meta-learning splitting.

| Meta-train | Meta-test | Art | Clipart | Product | Real_world | Ave. |
|---|---|---|---|---|---|---|
| $\mathcal{D}_s * 1$ | $\mathcal{D}_s * 2 + \mathcal{D}_{mix-up}$ | 70.36 | 58.23 | 80.40 | 83.99 | 73.23 |
| $\mathcal{D}_s * 2$ | $\mathcal{D}_s * 1 + \mathcal{D}_{mix-up}$ | 72.33 | 59.97 | 82.69 | 85.05 | 74.99 |
| $\mathcal{D}_s * 3$ | $\mathcal{D}_{mix-up}$ | 71.58 | 60.08 | 83.23 | 85.79 | 75.17 |

Table 9: mAP(%) on Office-Home with vision transformer feature extractor. Named in row is the target domain which contains 10 classes randomly selected from label space. (A: Art, C: Clipart, P: Product, R: Real_world)

| $\mathcal{D}_t^{10}$ | ERM-w/o | ERM-w | CCSA | MDAN | Mix-up | DAML | URT | Ours |
|---|---|---|---|---|---|---|---|---|
| A | 69.58 | 66.97 | 64.23 | 73.02 | 71.74 | 69.07 | 68.81 | 77.70 |
| C | 60.84 | 62.90 | 61.67 | 63.01 | 63.93 | 64.92 | 58.11 | 69.54 |
| P | 81.48 | 81.28 | 79.82 | 82.40 | 81.23 | 82.02 | 83.10 | 85.26 |
| R | 82.62 | 84.44 | 81.08 | 84.60 | 83.15 | 82.63 | 80.69 | 85.48 |
| Ave. | 73.63 | 73.61 | 71.70 | 76.01 | 74.90 | 74.66 | 75.18 | 79.94 |

## A.6   VISION TRANSFORMER FEATURE EXTRACTOR

We explore the performance of our method under different feature extractor by replacing the ResNet18 with ViT-B-16(vit_small_patch_224)[1] on Office-Home dataset with 10-shot. As shown in Table 9, we can see that our method still consistently outperforms all the baselines under the vision transformer feature extractor.

## A.7   MOVING DIRECTION OF MIX-UP DISTRIBUTION

We investigate the influence of moving direction of mix-up distribution on our P-Mixup by either moving the mix-up distribution from source to target domains ("Source-To-Target") or from target to source domains ("Target-To-Source"). In Table 10, we observe that the direction of "Target-To-Source" consistently and significantly outperforms the direction of "Source-To-Target" with more than $4.0\%$ improvement on average accuracy. To further explore the training behavior of our P-Mixup, we inspect the learned model from some of the intermediate training iterations, i.e., from iteration 500 to 5,000, to fully converged 10,000 iterations. As shown in Figure 6, We find that the direction of "Target-To-Source" continuously improves the performance of the learned model on target domain compared with the direction of "Source-To-Target".

## A.8   STANDARD DEVIATION VALUE

We compute the standard deviation (STD) values for all the methods across all the benchmark datasets. More details can be found in Figures 7, 8, and 9.

## A.9   COMPUTATION ANALYSIS

We explore our method training behavior by investigating the learned model from some of the intermediate training iterations, i.e., from iteration 2000 to 5000, to fully converged 10000 iterations. We run the models on Office-Home four protocols. As shown in Figure 5, we observe that our proposed method P-Mixup consistently and significantly outperforms all the baselines which verifies the effectiveness of our P-Mixup. We also notice that other baseline methods, e.g., DAML and MDAN, obtain worse results than EMR-w which demonstrates the difficulty and challenge of our proposed Few-shot Supervised multi-source domain Transfer (FSMDT) problem.

---

[1]https://timm.fast.ai/

Table 10: Impact of moving direction of mixup distribution in our P-Mixup on Office-Home (averaged over 5 times) all four protocols.

| Moving Direction | Art | Clipart | Product | Real_world | Ave. |
|---|---|---|---|---|---|
| Source-To-Target | 67.06 | 55.79 | 78.56 | 82.44 | 70.97 |
| Target-To-Source | 72.23 | 59.97 | 82.69 | 85.05 | 74.99 |

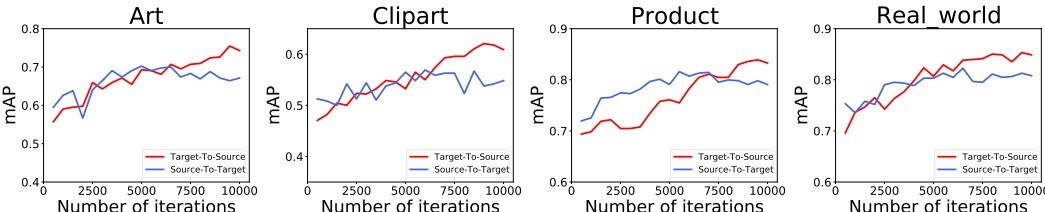

Figure 6: Illustration of our P-Mixup training behavior under different moving strategies for mix-up distribution on Office-Home (averaged over 5 times) all four protocols.

## A.10 LIMITATION

Our method mainly relies on source-target progressive mixup (P-Mixup) data augmentation, which progressively introduces an intermediate mix-up domain to mitigate the domain gap between source and target. P-Mixup focuses on the proposed Few-shot Supervised Multi-source Domain Transfer (FSMDT) problem, which provides multiple labeled source domain data and limited labeled target data. It aims at learning to generalize to unseen target domain data. To exactly constrain the target distribution to be few-shot, we consider one sample per class situation and limit the label space of target domain significantly smaller than the label space of multi-source domains. We can see in Table 4 that the performance of our method decreases as the number of target classes increases. There are two main reasons: First, the classification task becomes more difficult as the number of target classes increases. Second, the diversity of the augmented data is restricted by the fact that P-Mixup is only applied between source and the limited few-shot of target data. In contrast, the domain generalization method "DAML" conducts the mix-up across all the classes from multiple source domains. To mitigate the gap, we can additionally introduce the mix-up amongst source domains, e.g., source to source mix-up, into our overall framework to further enrich the data diversity.

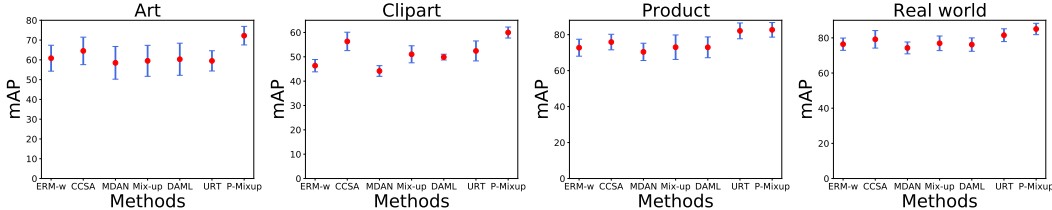

Figure 7: Illustration of standard deviation (STD) values for all the methods on Office-Home (averaged over 5 times) all four protocols.

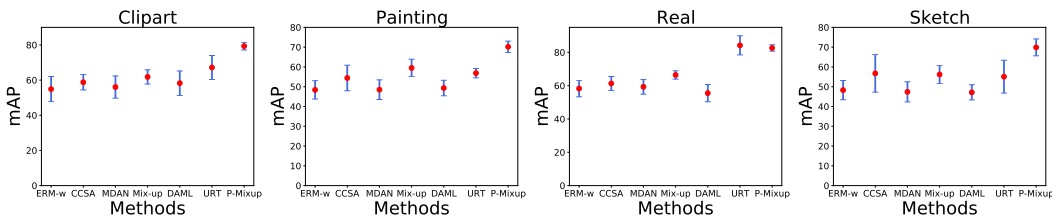

Figure 8: Illustration of standard deviation (STD) values for all the methods on DomainNet (averaged over 5 times) all four protocols, where target domain contains 10 classes.

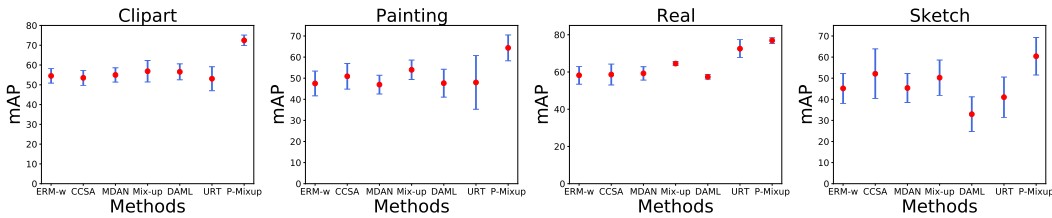

Figure 9: Illustration of standard deviation (STD) values for all the methods on DomainNet (averaged over 5 times) all four protocols, where target domain contains 15 classes.

