# OpenReview forum: "Progressive Mix-Up for Few-Shot Supervised Multi-Source Domain Transfer"
_ICLR.cc/2023/Conference — ICLR 2023 poster_

### Official Review · Reviewer_DEHW · 2022-10-23

**Confidence:** 4
**Clarity, Quality, Novelty And Reproducibility:** 1. The paper is the easy to read in m…
**Correctness:** 3
**Technical Novelty And Significance:** 3
**Empirical Novelty And Significance:** 3
**Recommendation:** 8

**Strength And Weaknesses:**

**Strengths :**

1. The paper is the easy to read in most cases.
2. The proposed method seems to make sense and it is easy to reproduce.
3. The experimental results look good and outperforms the baselines  with large margin.

**Weakness or question**
1. Sec 3.3 why choose the Wasserstein distance as the measurement ?  As far as I know, the the Wasserstein distance is hard to compute and how to compute it in your detail implement?

2. To the best of my knowledge, the proposed Few-shot Supervised Multi-source Domain Transfer (FSMDT) is similarity with the multi-domain few-shot learning \[1]\[2][3]   ,  I am very interested that the methods  \[1]\[2][3] designed for the multi-domain few-shot learning will work well in setting of FSMDT or not.

3. The performance improvement is significant. But could you design some more useful baselines? From my view, you used many weak baselines. So it is not a fair comparison.



[1] Liu L, Hamilton W L, Long G, et al. A Universal Representation Transformer Layer for Few-Shot Image Classification[C]//International Conference on Learning Representations. 2021.

[2] Dvornik N, Schmid C, Mairal J. Selecting relevant features from a multi-domain representation for few-shot classification[C]//European Conference on Computer Vision. Springer, Cham, 2020: 769-786.

[3] Triantafillou E, Larochelle H, Zemel R, et al. Learning a universal template for few-shot dataset generalization[C]//International Conference on Machine Learning. PMLR, 2021: 10424-10433.


**Summary Of The Paper:**

This paper propose a new setting named Few-shot Supervised Multi-source Domain Transfer (FSMDT),  which try to generalize from multiple source domains to a single target domain with few labeled samples. And  authors propose a novel method based on the mix-up trick to create an intermediate mix-up domain.  The experimental results show that the proposed framework outperforms the baselines in two datasets.


**Summary Of The Review:**

In this paper, the authors propose a new setting and a new method.  But I think there are still many details are unclear, or something importance reference and baselines is missing.

---

> ### Author Response · Authors · 2022-11-18
> **Response to Reviewer DEHW (Part 2)**
>
> **Q2.  I am very interested that the methods [1][2][3] designed for the multi-domain few-shot learning will work well in setting of FSMDT or not.**
>
> **A2.**   Thanks for the valuable suggestions. We have added Few-Shot Learning related works in our revised paper.  Among the three recommended methods [1, 2, 3], we pick the Universal Representation Transformer (URT)[1] as a new strong baseline in our setting, based on that URT[1] reports the best performance on multi-domain few-shot learning setting among [1, 2, 3]. The tables below show the comparison including a newly introduced dataset, i.e., PACS [4]  (we also added these results in our revised submission Table 1 and Table 2 of main submission and Table 7 in Appendix). We observe that URT works well and stands among the state-of-the-art methods. Even so, our method presents a clear improvement margin over the URT, the few-shot learning based framework.
>
>
> [1] ICLR2021: A Universal Representation Transformer Layer for Few-Shot Image Classification.
>
> [2] ECCV2020: Dvornik N, Schmid C, Mairal J. Selecting relevant features from a multi-domain representation for few-shot classification.
>
> [3] ICML2021: Triantafillou E, Larochelle H, Zemel R, et al. Learning a universal template for few-shot dataset generalization.
>
> [4] ICCV2017: Deeper, Broader and Artier Domain Generalization.
>
> |Office-Home with $\mathcal{D}_t^{10}$ | ERM-w/o |  ERM-w |  CCSA  | MDAN | Mix-up | DAML | URT | Ours |
> |----|:--------:|:--------:|:-------:|:--------:|:------:|:--------:|:--------:|:--------:|
> | A |  60.48 | 60.83 | 64.52 | 58.49 | 59.50| 60.30| 59.48| **72.23**|
> | C |  44.06 |46.35  | 56.32 |44.17  |51.03 | 49.88| 52.41| **59.97**|
> | P| 72.55  | 72.75 | 75.89 | 70.43 | 73.03| 72.96| 82.11| **82.69**|
> | R| 79.32  |76.40 | 79.17 | 74.28 |76.91 | 76.17| 81.52| **85.05**|
> | Ave.| 63.83  | 64.08| 68.97 | 61.84 | 65.12| 64.83| 68.88| **74.99** |
>
> |DomainNet with $\mathcal{D}_t^{10}$ | ERM-w/o |  ERM-w |  CCSA  | MDAN | Mix-up | DAML | URT | Ours |
> |----|:--------:|:--------:|:-------:|:--------:|:------:|:--------:|:--------:|:--------:|
> | C    |  47.33 | 50.28 | 43.89 | 52.33 | 56.83 | 57.05 | 63.93 | **65.06**|
> | P    |  37.52 | 43.36 | 41.28 | 43.86 | 52.35 | 47.95 | 56.02 | **60.85**|
> | R    | 47.66  | 58.34 | 46.46 | 58.02 | 61.60 | 58.88 | **72.91**| 70.60|
> | S    | 37.60  | 46.32 | 42.32 | 45.57 | 49.52 | 49.39 | 53.83 | **60.67**|
> | Ave.| 42.53  | 49.57 | 43.49 | 49.95 | 55.33 | 53.32 | 61.67 | **64.29** |
>
> |DomainNet with $\mathcal{D}_t^{15}$ | ERM-w/o |  ERM-w |  CCSA  | MDAN | Mix-up | DAML | URT | Ours |
> |----|:--------:|:--------:|:-------:|:--------:|:------:|:--------:|:--------:|:--------:|
> | C    |  45.74 | 49.77 | 41.57 | 51.34 | 52.98 | 54.54 | 52.21 | **63.92**|
> | P    |  42.43 | 45.16 | 42.76 | 46.61 | 51.90 | 46.81 | 47.38 | **58.37**|
> | R    | 53.03  | 56.46 | 47.89 | 57.02 | 58.94 | 58.67 | **69.31**| 67.15|
> | S    | 37.87  | 42.38 | 44.79 | 43.54 | 43.71 | 45.37 | 40.90 | **55.65**|
> | Ave.| 44.77  | 48.44 | 44.26 | 49.63 | 51.88 | 51.35 | 52.45 | **61.11** |
>
> |PACS with $\mathcal{D}_t^{4}$ | ERM-w/o |  ERM-w |  CCSA  | MDAN | Mix-up | DAML | URT | Ours |
> |----|:--------:|:--------:|:-------:|:--------:|:------:|:--------:|:--------:|:--------:|
> | A | 76.42 | 78.47 | 75.39 | 77.84 | 79.84 | 79.37 | 59.36 | **83.21**|
> | C | 70.15 | 72.37 | 74.04 | 70.15 | 71.57 | 76.86 | 62.61 | **79.96**|
> | P |  86.79 | 89.95 | 87.24 | 91.16 | 93.75 | 91.79 | 79.40 | **95.62**|
> | S|  64.98 | 69.22 | 73.29 | 70.10 | 66.32 | 72.79| 58.71| **81.97**|
> | Ave.|  74.58 | 77.50| 77.49 | 77.31 | 77.87| 80.20| 65.02| **85.19** |
>
> **Q3.  Could you design some more useful baselines?**
>
> **A3.**   Thanks for your valuable suggestion. For the current baselines, we have selected the representative methods from each of the sub-direction, e.g., Supervised Domain Adaptation, Multi-Source Domain Adaptation, Domain Generalization, and Data Augmentation. We also incorporated the SOTA Multi-Source Few-Shot Learning method, URT, into our comparison based on the reviewer’s valuable suggestion. Shown as in the table above, Our method shows consistently better results compared to those top performance baselines including URT.

---

> ### Author Response · Authors · 2022-11-18
> **Response to Reviewer DEHW (Part 1)**
>
> We thank the reviewer for providing constructive comments. In the following we provide detailed responses to these questions.
>
> **Q1.  why choose the Wasserstein distance as the measurement ? As far as I know, the the Wasserstein distance is hard to compute and how to compute it in your detail implement?**
>
> **A1.**  Wasserstein distance (W-distance) is a widely applied distance metric in measuring set-to-set distance. It shows large effectiveness in generative adversarial networks and domain adaptation/generalization. There are mature toolboxes to calculate the distribution to distribution distance. Below is the abstracted W-distance function we used in our experiments:
>
> we implement W-distance as:
>
> `compute_wd = SinkhornDistance(eps=0.1, max_iter=100)` \
> `wd, _, _ = compute_wd(s_feats, t_feats)`
>
> more details of the implement can be found in an anonymous dropbox <link>(https://www.dropbox.com/s/jfp44fufmy37a14/wdistance.zip?dl=0&file_subpath=%2Fwdistance.py).

---

> ### Comment · Reviewer_DEHW · 2022-11-21
> **To Response**
>
> Dear Authors,
>
> Thank you for your responses.
>
> I have read your rebuttal and I also have seen your efforts.
>
> My main concern is about the baselines and  the details of W-distance. In the rebuttal, you have done rich experiments to prove the efficiency of your method and you also give the link to introduce the details of W-distance. You have addressed all my concerns.
>
> Generally, few-shot learning is impossible to address without any other knowledge. In this paper, the authors introduce the knowledge from multiple source domains to improve the performance of few-shot target domain. This perspective is novel and makes sense. The paper also conduct reasonable experiments to prove that their method can handle this novel setting. From my view, this paper is the first paper to address this interesting and important setting.
>
> I agree to accept this paper. I will increase the score to accept.
>
> Best Regards.

---

> > ### Author Response · Authors · 2022-11-22
> > **Thank you for your valuable comments to improve our paper**
> >
> > Dear Reviewer,
> >
> > We sincerely thank you for your valuable comments, which helps us to improve the paper's demonstration, i.e., baselines and implementation details.
> >
> > Meanwhile, we fully acknowledge your thoughts on "few-shot learning is impossible to address without any other knowledge", and our work is indeed based on some hypothesis that "the variance or diversity within each source domain and the target domain shares high-level similarity." We utilize this to transfer the variance knowledge from source domains to the few-shot target domain.
> >
> > We further thank you for your kind acknowledgement of our submission "is the first paper to address this interesting and important setting". Hope our submission can bring an introduction to the research community for this important topic and welcome for further comments and discussions.
> >
> > Best regards,
> >
> > the authors

---

### Official Review · Reviewer_dGEs · 2022-10-24

**Confidence:** 3
**Clarity, Quality, Novelty And Reproducibility:** see the above section.
**Correctness:** 3
**Technical Novelty And Significance:** 3
**Empirical Novelty And Significance:** 3
**Recommendation:** 6

**Strength And Weaknesses:**

Pons:
This paper explores new setting which follows practical  application.

The paper gives clear description about this setting which is convincing.

This article is well-written and easy to follow.

The experiment results are sufficient  to support the proposed method.

I have a few concerns which are as following:

1) The proposed method is not vary suitable for multi-source domain adaptation. Why the mix-up technique works well for few-shot learning.  Since the number target dataset is small, the goal should locate the local information which is good for target domain. I just feel weird about the mix-up technique for few-shot learning.

2) Reproducibility is hard, since it refers more detail precessing about the proposed method. Also the code is not provided, thus I am not sure about reproducibility.

3) More datasets should be considered, since two datasets seem to be less.

4) It still works well if the network is transformer, do authors try it?



**Summary Of The Paper:**

This paper focuses on multi-source domain adaptation with limited target data.  Since in this setting there are few target data, current methods fail to learn efficient model to model it. In this paper, authors provide a progressive  mix-up method to learn it. The experiment results support the effectiveness of the proposed method.

**Summary Of The Review:**

I am not sure why the motivation why the mix-up is suitable for few-shot. Also the reproducibility should be challenge.

---

> ### Author Response · Authors · 2022-11-18
> **Response to Reviewer dGEs (Part 2)**
>
> **Q3.  More datasets should be considered, since two datasets seem to be less.**
>
> **A3.** Thanks for the suggestion. We adopt the PACS [R1] dataset as an additional third benchmark, and evaluate our method and baselines on it under the 4-shot setting. The new dataset and results have been added to our revised submission (Table 7 in Appendix). From the following table, we clearly see that our P-Mixup method shows the same trend as the two datasets reported in the original submission, which achieves consistently better accuracy against all the state-of-the-arts including a newly introduced multi-source few-shot learning method URT [R2].
>
> [R1] ICCV2017: Deeper, Broader and Artier Domain Generalization.
>
> [R2] ICLR2021: A Universal Representation Transformer Layer for Few-Shot Image Classification.
>
> |PACS with $\mathcal{D}_t^{4}$ | ERM-w/o |  ERM-w |  CCSA  | MDAN | Mix-up | DAML | URT | Ours |
> |----|:--------:|:--------:|:-------:|:--------:|:------:|:--------:|:--------:|:--------:|
> | A | 76.42 | 78.47 | 75.39 | 77.84 | 79.84 | 79.37 | 59.36 | **83.21**|
> | C | 70.15 | 72.37 | 74.04 | 70.15 | 71.57 | 76.86 | 62.61 | **79.96**|
> | P |  86.79 | 89.95 | 87.24 | 91.16 | 93.75 | 91.79 | 79.40 | **95.62**|
> | S|  64.98 | 69.22 | 73.29 | 70.10 | 66.32 | 72.79| 58.71| **81.97**|
> | Ave.|  74.58 | 77.50| 77.49 | 77.31 | 77.87| 80.20| 65.02| **85.19** |
>
>
> **Q4.  It still works well if the network is transformer, do authors try it?**
>
> **A4.**  Thanks for the valuable suggestion. We explore the performance of our method under different feature extractor by replacing the ResNet18 with Vision Transformer, i.e., ViT-B-32 (vit_small_patch32_224) from https://timm.fast.ai/,  on Office-Home dataset with 10-shot. As shown in Table below (we also add this experiment in our revised submission, Table 9 in Appendix), we observe the same trend that with the new ViT backbone, our method still consistently outperforms the other methods. While further comparing the ViT experiment table to Table 1 in main submission, we observe that the transformer based backbone shows ~5% performance improvement over the ResNet18 based backbone.
>
> |Office-Home with $\mathcal{D}_t^{10}$ | ERM-w/o |  ERM-w |  CCSA  | MDAN | Mix-up | DAML | URT | Ours |
> |----|:--------:|:--------:|:-------:|:--------:|:------:|:--------:|:--------:|:--------:|
> | A | 69.58  | 66.97 | 64.23 | 73.02 | 71.74 | 69.07 | 68.81 | **77.70**|
> | C | 60.84  | 62.90 | 61.67 | 63.01 | 63.93 | 64.92 | 58.11 | **69.54**|
> | P|  81.48 | 81.28 | 79.82 | 82.40 | 81.23 | 82.02 | 83.10 | **85.26**|
> | R|  82.62 | 84.44 | 81.08 | 84.60 | 83.15 | 82.63| 80.69| **85.48**|
> | Ave.|  73.63 | 73.61| 71.70 | 76.01  | 74.90 |74.66 | 75.18| **79.94** |

---

> ### Author Response · Authors · 2022-11-18
> **Response to Reviewer dGEs (Part 1)**
>
> We thank the reviewer for providing the constructive comments. In the following we provide detailed responses to these questions.
>
> **Q1. Why the mix-up technique works well for few-shot learning? The goal should locate the local information which is good for target domain.**
>
> **A1.**  We agree with the reviewer that “locate the local information which is good for the target domain” should be a promising idea. However, due to the very limited local information defined by the problem setting, i.e., only few or even one sample per class is observed, it is extremely hard to locate the local information.
>
> Instead, we make an assumption: '' the variance or diversity within each source domain and the target domain shares high-level similarity.'' Based on this assumption, though we only observe few shots from target domain, by exhaustively learning from the multiple sources and leveraging intermediate domains, we not only enrich the diversity of the target domain, but also gradually mitigate the domain gaps (i.e., target to mix-up domain, and source to mix-up domain). Eventually such processes effectively help to close the gap between the sources and the target.
>
> To empirically verify the above assumption that the target domain variance or diversity can be largely covered by source domains, we take the OfficeHome dataset and choose “Art” domain as the target domain. Assuming the center of the target domain is provided, we transfer the variance from the source domains onto the target center. Then, we calculate the Wasserstein distance between the transferred distribution and the target ground truth distribution. We also compute the Wassertein distance between the source distribution and the target ground truth distribution as a reference.  Here, we use imagenet pretrained ResNet18 as the feature extractor. We find that the transferred Wassertein distance is 0.6448 compared to the source-to-target Wassertein distance 0.8421.  The closer distance indicates that the transferred variance from sources is similar to the target variance and thus our assumption is indeed valid in this situation.
>
> **Q2. Reproducibility is hard. Also the code is not provided.**
>
> **A2.** We believe the technical details have been illustrated in the methodology and implementation details. To help better understand the designing, we put the P-Mixup core code under an anonymous dropbox <link>(https://www.dropbox.com/s/tgc5x76hv93ejuv/P-Mixup%5BCore_Code%5D.zip?dl=0&file_subpath=%2FP-Mixup%5BCore_Code%5D.py). Meanwhile, all the code will be released upon the paper acceptance.

---

> ### Comment · Reviewer_dGEs · 2022-11-29
> **Improving my score**
>
> Thanks for authors'comments. After carefully reading authors' feedback, I would like to improve my score, since it mitigates most of my questions.  To be specific, some of questions cost much of energy . Finally I hope authors release the code to boost this topic and help the researchers of  the community.

---

> > ### Author Response · Authors · 2022-12-02
> > **Thank you for your comments and acknowledgement**
> >
> > Dear Reviewer,
> >
> > We sincerely thank you for your time and efforts in providing us the valuable suggestions to improve this submission. As promised, we will release our code base upon the paper acceptance. We also would like to contribute our efforts as part to the community and welcome other researchers to explore into this area for better works.
> >
> > Best regards,
> >
> > The authors

---

### Official Review · Reviewer_Fa3w · 2022-10-24

**Confidence:** 3
**Correctness:** 3
**Technical Novelty And Significance:** 2
**Empirical Novelty And Significance:** 3
**Recommendation:** 6

**Clarity, Quality, Novelty And Reproducibility:**

This paper leverages mix-up for domain transfer, which can be viewed as an application extended from previous techniques. The written quality overall is high.

**Details Of Ethics Concerns:**

I have not found any ethics concerns.

**Strength And Weaknesses:**

Strength:

+ The research problem is important and may have many practical applications. In real-world machine learning applications, it is possible that the data from the target domain is limited. Improving the generalization of the pre-trained model on the source domain could be important and is a practical setting.

+ Extensive experiments are conducted.  The proposed method demonstrates strong performance and surpasses baselines by a large margin on two benchmark datasets.

+ This paper generally is well-written and easy to follow.

Weakness:

+ The technical insight may not be enough, the proposed method is mainly based on mix-up. Most of the motivation for using techniques is based on intuitive explanations. Transferring the knowledge to the target domain with very limited target data seems to be an ambitious task.  For me, it is not clear whether this method works in general or not. I think it is better to add some insightful analysis. For example, are there any assumptions to make the method useful? When will the proposed method fail to transfer the knowledge?
+ The related work under the proposed problem settings may not be fruitful. I am wondering that is it possible to compare one or two more baselines with minor tweaks in the related problem settings, (e.g., few-shot learning).


**Summary Of The Paper:**

This paper targets a research problem of how to transfer knowledge from multiple source domains to a single target domain with very limited target data. To solve the problem, a new method based on mix-up is proposed. It progressively pushes both the source domains and the few-shot target domain aligned to the mix-up domain.

**Summary Of The Review:**

This paper focuses on a practical and challenging domain transfer problem. Using mix-up for reducing transferring knowledge from the source domain to the target domain is an interesting method. My largest concern is that the technical insight may not that strong.

---

> ### Author Response · Authors · 2022-11-18
> **Response to Reviewer Fa3w (Part 2)**
>
> **Q2. Is it possible to compare one or two more baselines on few-shot learning?**
>
> **A2.** Thank you for the suggestion. We have adopted an additional state-of-the-art multi-source few-shot learning method, namely, Universal Representation Transformer (URT) [r1], as a new baseline. We evaluate the performance of URT in our proposed settings including a newly introduced dataset, i.e., PACS [r2], as shown in the following tables (We also reported these results in our revised submission, Table 1, 2 of main submission and Table 7 of appendix). We observe that our method still consistently outperforms URT and other methods across all the benchmarks with clear margin.
>
> [r1] ICLR2021: A Universal Representation Transformer Layer for Few-Shot Image Classification.
>
> [r2] ICCV2017: Deeper, Broader and Artier Domain Generalization.
>
> |Office-Home with $\mathcal{D}_t^{10}$ | ERM-w/o |  ERM-w |  CCSA  | MDAN | Mix-up | DAML | URT | Ours |
> |----|:--------:|:--------:|:-------:|:--------:|:------:|:--------:|:--------:|:--------:|
> | A |  60.48 | 60.83 | 64.52 | 58.49 | 59.50| 60.30| 59.48| **72.23**|
> | C |  44.06 |46.35  | 56.32 |44.17  |51.03 | 49.88| 52.41| **59.97**|
> | P| 72.55  | 72.75 | 75.89 | 70.43 | 73.03| 72.96| 82.11| **82.69**|
> | R| 79.32  |76.40 | 79.17 | 74.28 |76.91 | 76.17| 81.52| **85.05**|
> | Ave.| 63.83  | 64.08| 68.97 | 61.84 | 65.12| 64.83| 68.88| **74.99** |
>
> |DomainNet with $\mathcal{D}_t^{10}$ | ERM-w/o |  ERM-w |  CCSA  | MDAN | Mix-up | DAML | URT | Ours |
> |----|:--------:|:--------:|:-------:|:--------:|:------:|:--------:|:--------:|:--------:|
> | C    |  47.33 | 50.28 | 43.89 | 52.33 | 56.83 | 57.05 | 63.93 | **65.06**|
> | P    |  37.52 | 43.36 | 41.28 | 43.86 | 52.35 | 47.95 | 56.02 | **60.85**|
> | R    | 47.66  | 58.34 | 46.46 | 58.02 | 61.60 | 58.88 | **72.91**| 70.60|
> | S    | 37.60  | 46.32 | 42.32 | 45.57 | 49.52 | 49.39 | 53.83 | **60.67**|
> | Ave.| 42.53  | 49.57 | 43.49 | 49.95 | 55.33 | 53.32 | 61.67 | **64.29** |
>
> |DomainNet with $\mathcal{D}_t^{15}$ | ERM-w/o |  ERM-w |  CCSA  | MDAN | Mix-up | DAML | URT | Ours |
> |----|:--------:|:--------:|:-------:|:--------:|:------:|:--------:|:--------:|:--------:|
> | C    |  45.74 | 49.77 | 41.57 | 51.34 | 52.98 | 54.54 | 52.21 | **63.92**|
> | P    |  42.43 | 45.16 | 42.76 | 46.61 | 51.90 | 46.81 | 47.38 | **58.37**|
> | R    | 53.03  | 56.46 | 47.89 | 57.02 | 58.94 | 58.67 | **69.31**| 67.15|
> | S    | 37.87  | 42.38 | 44.79 | 43.54 | 43.71 | 45.37 | 40.90 | **55.65**|
> | Ave.| 44.77  | 48.44 | 44.26 | 49.63 | 51.88 | 51.35 | 52.45 | **61.11** |
>
> |PACS with $\mathcal{D}_t^{4}$ | ERM-w/o |  ERM-w |  CCSA  | MDAN | Mix-up | DAML | URT | Ours |
> |----|:--------:|:--------:|:-------:|:--------:|:------:|:--------:|:--------:|:--------:|
> | A | 76.42 | 78.47 | 75.39 | 77.84 | 79.84 | 79.37 | 59.36 | **83.21**|
> | C | 70.15 | 72.37 | 74.04 | 70.15 | 71.57 | 76.86 | 62.61 | **79.96**|
> | P |  86.79 | 89.95 | 87.24 | 91.16 | 93.75 | 91.79 | 79.40 | **95.62**|
> | S|  64.98 | 69.22 | 73.29 | 70.10 | 66.32 | 72.79| 58.71| **81.97**|
> | Ave.|  74.58 | 77.50| 77.49 | 77.31 | 77.87| 80.20| 65.02| **85.19** |

---

> ### Author Response · Authors · 2022-11-18
> **Response to Reviewer Fa3w (Part 1)**
>
> We thank the reviewer for providing the constructive comments. In the following we provide detailed responses to these questions.
>
> **Q1. Transferring the knowledge to the target domain with very limited target data seems to be an ambitious task. It is not clear whether this method works in general or not?**
>
> **A1.** Our contribution lies in closing the domain gap from sources to a few-shot observed target domain, where mix-up is utilized as a tool to form the intermediate domain. We agree that transferring knowledge to the target domain with few-shots is an ambitious task. To make it feasible, we have the following assumption: ''the variance or diversity within each source domain and the target domain shares high-level similarity. ''
>
> Based on this assumption, though we only observe few shots from target domain, by exhaustively learning from the multiple sources and leveraging the intermediate domain, we not only enrich the diversity of the target domain, but also gradually mitigate the domain gaps (i.e., target to mix-up domain, and source to mix-up domain). Eventually such processes effectively help close the gap between the sources and the target.
>
> To empirically verify the above assumption that the target domain variance or diversity can be largely covered by source domains, we take the OfficeHome dataset and choose “Art” domain as the target domain. Assuming the center of the target domain is provided, we transfer the variance from the source domains onto the target center. Then, we calculate the Wasserstein distance between the transferred distribution and the target ground truth distribution. We also compute the Wassertein distance between the source distribution and the target ground truth distribution as a reference.  Here, we use imagenet pretrained ResNet18 as the feature extractor. We find that the transferred Wassertein distance is 0.6448 compared to the source-to-target Wassertein distance 0.8421.  The closer distance indicates that the transferred variance from sources is similar to the target variance and thus our assumption is indeed valid in this situation.
>
> For the failure cases, it is also clear that: if our hypothesis does no longer hold, i.e., if there exists some situation, where the ground truth target variance is largely different from the multiple sources' variance, it is likely to fail. Here variance refers to within-class variance, not the target distribution.

---

> > ### Comment · Reviewer_Fa3w · 2022-11-28
> > **My concerns have been addressed.**
> >
> > Thank you for the clarification. My major concerns above novelty and assumptions required by this paper have been addressed. The assumption has been clearly justified. I prefer to accept this paper and keep my original rate.
> >
> >
> > Reviewer Fa3w

---

> > > ### Author Response · Authors · 2022-11-28
> > > **Thank you for your valuable comments and acknowledgement**
> > >
> > > Dear Reviewer Fa3w,
> > >
> > > We sincerely thank you for the time and efforts in helping us improve the quality of our submission, i.e., the novelty clarification and the motivation assumption. We also thank you for your further acknowledgement on the merit of the research submission. Any further comment or discussion is welcomed.
> > >
> > > Best regards,
> > >
> > > the authors

---

### Official Review · Reviewer_6uuP · 2022-11-05

**Confidence:** 4
**Correctness:** 3
**Technical Novelty And Significance:** 3
**Empirical Novelty And Significance:** 2
**Recommendation:** 6

**Clarity, Quality, Novelty And Reproducibility:**

The work introduced a new setting of FSMDA which hadn't been proposed before.  Under this new setting, they proposed a simple progressive mix-up method that outperforms the SOTA on two benchmarks, which has enough novelty.

**Details Of Ethics Concerns:**

nil

**Strength And Weaknesses:**

Strength:

The work introduced a new setting of domain adaptation with a few labeled target samples and labeled multiple source domain data which hadn't been proposed before. The proposed setting is useful in practice. They proposed a  simple progressive mix-up scheme that outperforms the current SOTA on the two benchmarks. The paper is basically well-written and easy to follow.

Weaknesses:
- I have some concerns about the method. And, some places of the method are not very clear.
   - The progressive mix-up formulation of $\lambda$ in EQ8: there is no grantee that $\lambda$ is in the range of [0 1]. According to EQ8, the final value of $\lambda_N = \lambda_{N-1}$ with $q=0$, where the q ranges from $\frac{1}{e}$.
   - In EQ6 and 4, the method uses mixed-up multi-class regression labels. It's not clear in the paper whether they linearly combine the labels across the classes or they only mix the labels across the domains with the same class. Few-shot samples from more classes should provide a more diversified and complete target domain distribution. However, the experiments show the method doesn't get better results with more few-shot samples. What is the reason?

- The experiments are not very strong with a few concerns.
   - The paper only provided their results on two benchmarks. Usually, more results are required.
   - The few-shot results with all the class samples are required. It only shows the results with the samples from 10~25 classes.  It is not clear whether the method can deal with a big class number (i.e. will the performance deteriorate with a big class number ?)
   - More ablation results and discussions about the meta-training process are required, e.g. the split selection of 60%source domains only for meta- train + 40% all domains for meta-test.



**Summary Of The Paper:**

This work introduced a new setting of few-shot supervised mulit-source domain adaptation with a few labeled target samples and labeled multiple source domain data. Under this new setting, they proposed a progressive mix-up method for few-shot supervised multi-source domain transfer. Specifically, it creates an intermediate mix-up domain and gradually updates the mix-up ratio $\lambda$ to mitigate the domain between the target domain and the source domain. They followed MAML(Finn 2017) to train the model. Their experiments show that the proposed method can outperform the previous methods on the two benchmarks of "Office-home" and "DomainNet".

**Summary Of The Review:**

In summary, they proposed a progressive mix-up method with a new setting of FSMDA. However, there are some concerns about the proposed method and the experimental results.

---

> ### Author Response · Authors · 2022-11-19
> **Response to Reviewer 6uuP (Part 2)**
>
>
> **Q3. The paper only provided their results on two benchmarks. Usually, more results are required.**
>
> **A3.**  As the reviewer raised, besides the two commonly applied benchmarks in our main submission, we have added another dataset, the PACS [R1] dataset, and evaluated our method and baselines on it under the 4-shot setting. The new results and discussions have been added to our revised paper (Table 7 in Appendix). From the table below, we clearly see that our P-Mixup method shows the same trend as the two datasets reported in our submission, which achieves consistently better results against all the state-of-the-arts including a newly introduced multi-source few-shot learning method URT [R2].
>
> [R1] ICCV2017: Deeper, Broader and Artier Domain Generalization.
>
> [R2] ICLR2021: A Universal Representation Transformer Layer for Few-Shot Image Classification.
>
> |PACS with $\mathcal{D}_t^{4}$ | ERM-w/o |  ERM-w |  CCSA  | MDAN | Mix-up | DAML | URT | Ours |
> |----|:--------:|:--------:|:-------:|:--------:|:------:|:--------:|:--------:|:--------:|
> | A | 76.42 | 78.47 | 75.39 | 77.84 | 79.84 | 79.37 | 59.36 | **83.21**|
> | C | 70.15 | 72.37 | 74.04 | 70.15 | 71.57 | 76.86 | 62.61 | **79.96**|
> | P |  86.79 | 89.95 | 87.24 | 91.16 | 93.75 | 91.79 | 79.40 | **95.62**|
> | S|  64.98 | 69.22 | 73.29 | 70.10 | 66.32 | 72.79| 58.71| **81.97**|
> | Ave.|  74.58 | 77.50| 77.49 | 77.31 | 77.87| 80.20| 65.02| **85.19** |
>
> **Q4. The few-shot results with all the class samples are required. It only shows the results with the samples from 10~25 classes. It is not clear whether the method can deal with a big class number.**
>
> **A4.** Thanks for the valuable comments. Based on the Table 4 in main submission, we further evaluate our method and baselines with 100 classes and 345 classes settings on DomainNet dataset (We have updated the Table 4 in main submission). From the table below, we observe that our method can successfully deal with the situations with large number of classes, i.e., 100 or 345 classes on DomainNet without significant performance degradation.
>
> |$\mathcal{D}_t^{n}$ | ERM-w/o |  ERM-w |  CCSA  | MDAN | Mix-up | DAML | URT | Ours |
> |----|:--------:|:--------:|:-------:|:--------:|:------:|:--------:|:--------:|:--------:|
> |10  |47.33 |50.28 |43.89 |52.33 |56.83 |57.15 |63.93 |65.06  |
> | 15  |45.74 |49.77 |41.57 |51.34 |52.98 |54.45 |52.21 |63.92 |
> | 20  |47.04 |51.65 |42.54 |53.07 |52.11 |56.04 | 53.67 |66.26|
> | 25  |47.93 |52.89 |45.43 |54.55 |50.54 |57.82 | 59.03 |65.82|
> | 100 |52.65 |54.25 |51.98 |58.74 |46.69 |60.41 |28.27 |62.88|
> | 345 |50.58 | 52.89 |48.73 |57.47 | 46.19 |58.77 |20.48 |63.63 |
>
> **Q5. More ablation results and discussions about the meta-training process are required, e.g. the split selection of 60% source domains only for meta-train + 40% all domains for meta-test.**
>
> **A5.**  Thanks for the valuable comments. We investigate the behavior of our method on different meta-train and meta-test splittings on Office-Home with 10-shot. As the Office-Home dataset contains 4 domains, for each task, there are 3 domains selected as the source domains and the remaining is the target domain. We also have the mix-up domain in each task. Due to the limited target data, we simplify the splitting by treating the target and mix-up domains as a whole, denoted as $\mathcal{D}_{t}$. From the table below, we increase the size of meta-train set from 1 source domain to 3 source domains. Correspondingly, the remaining domains are adopted as the meta-test set. We find that different meta-learning splittings achieve similar performance, and the meta-train set with three source domains slightly outperforms others (We also reported these results in our revised submission, Table 8 of appendix )
>
> |Meta-Train | Meta-Test  |  A  | C |  P  |  R  |  Ave.  |
> |----|:--------:|:--------:|:-------:|:--------:|:------:|:--------:|
> |$\mathcal{D}_s$*1| $\mathcal{D}_s$*2, $\mathcal{D}_t$ |70.36 |58.23 | 80.40| 83.99| 73.25|
> |$\mathcal{D}_s$*2|  $\mathcal{D}_s$*1, $\mathcal{D}_t$ |72.33 | 59.97 | 82.69 | 85.05 | 74.99|
> |$\mathcal{D}_s$*3| $\mathcal{D}_t$ | 71.58 | 60.08 | 83.23 | 85.79 | 75.17 |

---

> > ### Comment · Reviewer_6uuP · 2022-11-28
> > **Response to the revision**
> >
> > Thanks for the clarification. My major concern in Q4 hasn't been addressed. The proposed method may have problems to deal with a big class number. The authors only show the results with the target samples from 10~25 classes in the original version.  When the class number increases (100&345 in the revision), the performance drops. Intuitively, DA performance should increase with more labeled samples since k-shot samples from more classes provide a more diversified and complete target domain distribution. The same problem also can be found in the Mix-up and URT methods. The reason is not clear, the authors haven't mentioned the problem or discussed the limitations in the paper. I prefer to keep my original rate.

---

> > > ### Author Response · Authors · 2022-11-28
> > > **Further clarification to Reviewer 6uup's comments**
> > >
> > > We sincerely thank the reviewer’s valuable feedback. We would like to make some further clarification based on the reviewer’s comments.
> > >
> > > **Reviewer: "The proposed method cannot deal with a big class number".**
> > >
> > > **Answer:** We respectfully disagree because in A4, our method significantly and consistently achieved clear better results compared to other state-of-the-art (SOTA) methods, on both small class number (10-25) and big class number (100, 345), e.g., on 345 class, ours is 23.15% better than URT, 4.86% better than DAML, 17.44% better than Mix-up, 10.74% better than ERM, etc.
> > >
> > > We believe the reviewer might ask for the following (please correct us if we are wrong):
> > >
> > > **Reviewer:  with class number 100 or 345, why it is worse than class number 10,15, or 25 within our own method in A4 table.**
> > >
> > > **Answer:** We hereby further clarify:
> > >
> > > From the reviewer’s statement, “intuitively, DA performance should increase with more labeled target samples.” We are sorry if we do not put it clear enough with the efforts in Q2&A2 together. The main reasons are:
> > >
> > > 1. The setting in A4 table is exactly “one-shot”. In A2, we have shown that, in this extreme setting (one-shot), it does not introduce “more variance” for each class, but rather only increases the class number. Thus we believe in this case, it is not necessary to increase the DA performance.
> > > 2. During Testing, those 10,15 or 25 class experiments are only conducting 10-way, 15-way or 25-way classification task respectively. While for 100 or 345 classes, it conducts a 100-way or 345-way class classification problem, which is clearly a harder task.
> > >
> > > We believe the statement from the reviewer is valid based on a different setting (not one-shot setting), which is in Q2&A2. We set the number of images per class to be 20 (not one-shot), and increase the number of classes from 50, 60 to 70 and 80, which we indeed observe the performance improvement.
> > >
> > > **Discussions**: in brief, this suggests that: with the number of classes increasing, the performance continuously increasing is “Conditional”. When in the situation with extreme one-shot, it is not strictly to be monotonically increasing. When in the situation with sufficient samples per class (e.g., 20), it is more towards monotonically increasing.
> > >
> > > Thanks again for the reviewer’s constructive analysis. Hope our further clarification helps to alleviate the concern. We will add the above discussion into the paper final version.
> > >
> > > Best regards,
> > >
> > > the authors

---

> > > > ### Comment · Reviewer_6uuP · 2022-11-30
> > > > **My concerns about the paper**
> > > >
> > > > My main concerns to the paper are the experimental setting and results . Most of DA methods evaluate the preformance of the whole classes in the datasets. However, this paper proposed to just $\textbf{evaluate a subset of classes}$ which is $\textit{rare}$ in the previous literatures.  That was the reason I concerned whether the method could deal with large class number well, and asked for additional results. The additional results show that the proposed method may have problems to deal with large number classes.
> > > >
> > > > The authors argued that it was "conditionally" true that the DA performan"continuously" increases with inceasing labelled samples. The performance is not necessarily "continously" increasing, but more labelled target samples should help the domain alignment even when the samples are biased. And, K-shot samples are actually uniformlly sampled from each class.  As shown in A4, vanilla ERM with more samples gets better results as well as other methods of CCSA, MDAN, DAML. However, both the 100&345 results of the method drop by 3~4% from the best case.
> > > >
> > > > The proposed method may have problems when the class number increases as URT and Mixup. The authors should either
> > > > -  provide experiment resutls to show the method has not such a problem.
> > > >
> > > > or
> > > >
> > > > - provide the discussion and analysis about the problem and its possible reasons.
> > > >
> > > > but not  propose an experiment setting to evaluate small class subsets as in the original submittion, or avoid to discuss the limitations.
> > > >
> > > > In summary, I've not been convinced by the current results, and would like keep the original rate.

---

> > > > > ### Author Response · Authors · 2022-12-02
> > > > > **Further clarification to Reviewer 6uup's comments**
> > > > >
> > > > > We thank reviewer 6uuP for the very insightful comments. We re-clarify the concern from the reviewer as the following:
> > > > >
> > > > > - Actually, our evaluation is on the few-shot subset classes classification, not on the large number classes or the full set of the classes.
> > > > >
> > > > > - Based on the results in Table A4, as the reviewer mentioned, for each method, if compared to each of its own, when the number of classes increases, the performance of our method stays at the same level, not further increasing.
> > > > >
> > > > > - We apologize if by any means we made the misunderstanding that we will skip such discussions on method limitation. We hereby provide our thoughts and hope it makes sense. The following discussions on limitations of our method will be added to our final version.
> > > > >
> > > > > - The general ERM and domain adaptation methods such as MDAN show generally better results when the number of classes increases. This is because, as the reviewer mentioned, the more classes introduced, it better helps domain alignment.
> > > > >
> > > > > **Limitation of our method:**
> > > > >
> > > > > However, this does not hold for Mix-up, URT and our method. Due to the few-shot constraint, when conducting mix-up, the extremely partial variance in target domain is actually propagated to the intermediate mixed-up domain, which is possibly to amplify the distribution bias. By further pushing the source and target to be close to the intermediate domain, the alignment could be sub-optimal.
> > > > >
> > > > > **Advantage to other SOTA methods:**
> > > > >
> > > > > Even so, our method is still clearly better than Domain Adaptation methods (e.g., MDAN), 63.63% compared to 57.47% when class number is the full set of 345 classes. We believe this is because, when directly conducting the domain adaptation, the few-shot target domain provides very limited distribution variance, and thus by its own it is hard to align well with the source domains. While for our method, though there is a possible distribution bias propagated to the mixed-up domain, we are leveraging the fully observable source domains’ variances and transferring it to the few-shot target domain, thus enriching the target domain variances. With the enriched target domain variances, it benefits our method to do a better job than the traditional DA methods.
> > > > >
> > > > > Thanks again for your constructive comments and valuable suggestions! They are very helpful for improving our paper. We hope our clarifications help alleviate your concern, and we are more than happy to provide additional information if you have further questions.

---

> > > > > > ### Comment · Reviewer_6uuP · 2022-12-02
> > > > > > **Responce to the further clarification**
> > > > > >
> > > > > > Thank the authors for the further clarification. Subset evaluation is only valid for few-shot learning (for new classes). The proposed method is not a few-shot learning method for learning new classes but uses the few-shot constraint to solve the DA problem. For domain alignments/adaptation/generation problems, a whole domain evaluation is required. I suggest that the paper provides the evaluation results of the whole target domains instead of only small subsets. If not, the authors should provide previous literature in DA which uses similar experimental settings to just evaluate the target subsets.
> > > > > >
> > > > > > Also, the discussion without analysis about why the performance drops with more constraints from the target domain is too general and brief.

---

> > > > > > > ### Author Response · Authors · 2022-12-05
> > > > > > > **reply to reviewer's further feedback**
> > > > > > >
> > > > > > > We thank the reviewer for the further comments and clarification. Firstly, we agree with the reviewer that putting up the evaluation on the full set of classes is valid. Since it is a completely different setting, it takes time to conduct the whole experiments. We are actively running the experiments and will update the results within the discussion period. Even if it does not catch the discussion period due to  the experiments running time, we promise to incorporate the full set class experiment across the three datasets in our final version.
> > > > > > >
> > > > > > > To answer another question from the reviewer of whether there are methods that exactly adopt the partial set of classes for evaluation, we indeed find a research direction, which is named “Partial Domain Adaptation”. These methods consider leveraging the partially observable target data and together with the source domain data for the domain adaptation. Below we listed the most recent top tier conference papers for support. Moreover, we quickly pick one of them that achieves top performance and compared to our method on PACS, where our method shows certain advantage over the PDA method ETN[3].
> > > > > > > |Method |  A   |  C  |  P  |  S  | Avg. |
> > > > > > > |:------:|:------:|:------:|:--------:|:--------:|:--------:|
> > > > > > > |ETN[3]|76.3|67.5|87.0|62.6 | 73.4|
> > > > > > > |Ours | 83.2 | 80.0 | 95.6 | 82.0 | 85.2 |
> > > > > > >
> > > > > > > [1] Z. Cao, L. Ma, M. Long, and J. Wang. Partial adversarial domain adaptation. ECCV, 2018.
> > > > > > >
> > > > > > > [2] J. Zhang, Z. Ding, W. Li, and P. Ogunbona. Importance weighted adversarial nets for partial domain adaptation. CVPR, 2018
> > > > > > >
> > > > > > > [3] Z. Cao, K. You, M. Long, J. Wang, Q. Yang. Learning to transfer examples for partial domain adaptation. CVPR, 2019.
> > > > > > >
> > > > > > > [4] J. Liang, Y. Wang, D. Hu, R. He, and J. Feng. A balanced and uncertainty-aware approach for partial domain adaptation. ECCV, 2020.
> > > > > > >
> > > > > > > [5] Z. Chen, C. Chen, Z. Cheng, B. Jiang, K. Fang, and X. Jin. Selective transfer with reinforced transfer network for partial domain adaptation. CVPR, 2020.
> > > > > > >
> > > > > > > [6] W. Xiao, Z. Ding, and H. Liu. Implicit semantic response alignment for partial domain adaptation. NeurIPS, 2021
> > > > > > >
> > > > > > > [7] X. Gu, X. Yu, Y. Yang, J. Sun. and Z. Xu. Adversarial reweighting for partial domain adaptation. NeurIPS, 2021.
> > > > > > >
> > > > > > > [8] Y. Kim, S. Hong, S. Yang, S. Kang, Y. Jeon, and J. Kim. Associative Partial Domain Adaptation. AAAI, 2021.

---

> ### Author Response · Authors · 2022-11-19
> **Response to Reviewer 6uuP (Part 1)**
>
> We thank the reviewer for providing constructive comments. In the following we provide detailed responses to these questions.
>
> **Q1.  there is no guarantee that λ is in the range of [0 1], according to EQ8**
>
> **A1.** The reviewer had a good catch for Eqn. 8. While that is true, we further introduced Eqn. 9, which is a “clamp” operation to guarantee λ is in the range of [0, 1].
>
> **Q2.It's not clear in the paper whether they linearly combine the labels across the classes or they only mix the labels across the domains with the same class. the experiments show the method doesn't get better results with more few-shot samples. What is the reason?**
>
> **A2.** To clarify, we mix the labels across both domains and classes, where the labels across the domains are not necessary to be within the same class.
>
> While we agree with the high-level statement that “more classes should provide a more diversified and complete target domain distribution”, this conclusion is conditionally true. First, there is a point we need to clarify based on our proposed setting. The definition of “shot” in our setting is actually equal to the number of target classes since we only have one sample per class, and the label space of target test data is equal to the number of “shot” instead of source label space. As a result, increasing few-shot samples means increasing the size of target label space, but each class still only has one sample which cannot enrich the within-class distribution. For example, in 10-shot and 50-shot settings, even though we have more samples in 50-shot setting compared with 10-shot setting, the classification tasks for both settings are totally different, i.e., the 10-shot setting only considers the 10-way classification task while the latter one has to deal with the 50-way classification task, which is more difficult. Thus, we find that methods in our experiment do not achieve better results with more “shot” samples in Table 4 of main submission, since more “shot” samples do not bring more within-class information to each class, but just simply introduces new classes.
>
> To further verify “more classes should provide a more diversified and complete target domain distribution”, we redesign an experimental setting by increasing the number of target classes from 50 to 80 while keeping each class containing 20 labeled samples (not only 1 sample), the label space of target test data is fixed to 50 classes. We evaluate the performance of “EMR-w” on this redesigned setting. As shown in the table below, we observe the performance increasing trend as the reviewer mentioned.
>
> |  50-class  | 60-class | 70-class | 80-clas |
> |:------:|:--------:|:--------:|:--------:|
> |58.19|59.45|60.74|64.99|
>
> Putting the above two folds together, we see that the reviewer’s statement is conditionally true, if the number of shots per class is sufficient, e.g., 20 labeled samples per class. If the number of shots is one or very few, increasing the number of classes does not necessarily increase the within-class information and thus not improving performance.

---

### Decision · Program_Chairs · 2023-01-20

**Decision:**

Accept: poster

**Justification For Why Not Higher Score:**

What didn't get this work a higher score was insufficient technical insights as mentioned in the meta review.

**Justification For Why Not Lower Score:**

This paper provides a new problem setting of domain adaption. Although the proposed method is mainly based upon the prior work (i.e., Mixup), the method works well under this new setting, which clearly outperforms baselines. As this paper can contribute to the research community and inspire follow-up research, AC chooses "Accept with poster".

**Metareview: Summary, Strengths And Weaknesses:**

This paper presents work on multi-source domain adaptation with limited target data. Considering that there are a few target data, prior works fail to learn effective models for domain adaptation. To handle the problem, a mixup-based method is proposed, where a mix-up domain is produced. Experiments on benchmarks confirm the effectiveness of the proposed method.

The strengths of this paper include (a) The introduced problem setting of domain adaptation is practical and has potential. In real-world scenes, the data of the target domain may be limited. (b) The experiments in this paper look great. The proposed method outperforms the baselines by a large margin. (c) The writing of this paper is overall great. The weaknesses of this paper mainly lie in the technical insight may be insufficient. The proposed method highly relies on the Mixup method. Besides, the rationality of exploited techniques is based on intuitive explanations.

Before rebuttal, four reviewers post several concerns on this paper, including the discussions about related work, the characterizations of the proposed method, and questions about partial experiments. The authors provide detailed responses and actively communicate with reviewers. Reviewers are satisfied with their responses and the revised version of this paper. They hence tend to accept it. AC checks all activities in the review process and this paper, and agrees with the reviewers. Therefore, the recommendation is "accept".


**Note From Pc:**

if the above contains the word "oral" or "spotlight" please see: "oral" presentation means -> notable-top-5% and "spotlight" means -> notable-top-25%. As stated in our emails, we are disassociating presentation type from AC recommendations

**Summary Of Ac-Reviewer Meeting:**

AC and reviewers have reached a consensus in the internal discussion.